# Structural basis for sarbecovirus Rc-o319 spike adaptation to *Rhinolophus cornutus* Bat ACE2 and constraints on switching to human ACE2

Jingjing Wang[1,2☉], Zexuan Li[1,3☉], Yong Ma[1,2☉], Zimu Li[1,2,4], Hang Yuan[1,2,3], Chuanying Niu[1,5], Yifeng Teng[1,3], Banghui Liu[1,2], Mei Li[4], Min Zhou[4], Wenxiu Liu[4], Huimin Feng[1], Jing Chen[4], Jun He[1,2], Xinwen Chen[4,6]*, Xiaoli Xiong [1,2,4]*

1 State Key Laboratory of Respiratory Disease, Guangdong Provincial Key Laboratory of Stem Cell and Regenerative Medicine, Guangdong-Hong Kong Joint Laboratory for Stem Cell and Regenerative Medicine, Guangzhou Institutes of Biomedicine and Health, Chinese Academy of Sciences, Guangzhou, China, 2 Guangzhou Medical University-Guangzhou Institutes of Biomedicine and Health Joint School of Life Sciences, Guangzhou Medical University, Guangzhou, China, 3 University of Chinese Academy of Sciences, Beijing, China, 4 Guangzhou National Laboratory, Guangzhou, China, 5 University of Science and Technology of China, Hefei, China, 6 State Key Laboratory of Respiratory Disease, The First Affiliated Hospital of Guangzhou Medical University, Guangzhou, China

☉ These authors contributed equally to this work.
* chen_xinwen@gzlab.ac.cn (XC); xiong_xiaoli@gzlab.ac.cn (XX)

## Abstract

Bat sarbecoviruses often exhibit species-dependent ACE2 specificity. Understanding the determinants of receptor specificity enables better assessment of the cross-species transmission potential of sarbecoviruses. Here, we characterize the S-protein of Rc-o319, a sarbecovirus identified in Japanese *Rhinolophus cornutus* bats. Featuring an unusual 9-amino-acid deletion within its receptor binding motif (RBM), Rc-o319 S-protein utilizes its cognate *R. cornutus* ACE2 (bACE2$_{R.cor}$) but not human ACE2 (hACE2), demonstrating highly restricted receptor specificity. Cryo-EM structures reveal two locked prefusion conformations of the Rc-o319 S-trimer and define a novel type of receptor-binding domain (RBD), featuring a distinct beta-loop (BL) within the RBM due to the RBM-deletion. The Rc-o319-RBD:bACE2$_{R.cor}$ complex structure reveals unique interactions mediated by the specialized BL and RBM-loop of Rc-o319-RBD and by a bACE2$_{R.cor}$ glycan. Structure-guided mutagenesis demonstrates that changes in BL and RBM-loop within the Rc-o319 S-RBD must occur simultaneously to allow medium-to-high-affinity hACE2 binding. Comparative assays further show that the bACE2$_{R.cor}$ receptor supports only a subset of sarbecoviruses, highlighting its restricted sarbecovirus compatibility. Our findings establish the Rc-o319 S-protein as a structurally and functionally specialized adaptation to *R. cornutus* ACE2 and identify the structural constraints limiting its cross-species transmission potential.

**Data availability statement:** The Cryo-EM density maps and coordinates generated in this study have been deposited in the Electron Microscopy Data Bank (EMDB; www.ebi.ac.uk/emdb/) and the Protein Data Bank (PDB; www.rcsb.org/) with the following accession numbers: Rc-o319 locked-1 S-trimer: EMD-65717 and PDB: 9W75; Rc-o319 locked-2 S-trimer: EMD-65718 and PDB: 9W76; Dimeric Rc-o319-RBD:bACE2$_{R.cor}$ complex: EMD-65719 and PDB: 9W77, monomeric Rc-o319-RBD:bACE2$_{R.cor}$ complex: EMD-65720 and PDB: 9W78. All data needed to evaluate the conclusions in the paper are present in the paper and/or the Supplementary Materials.

**Funding:** This work was supported by the National Natural Science Foundation of China (32570199 and 82341085 to X.X.); the National Key R&D Program of China (2021YFA1300903 to X.X.); the Major Project of Guangzhou National Laboratory (SRPG22-002 to X.X.; SRPG22-003 to J.H.); the Guangdong Basic and Applied Basic Research Foundation (2021A1515011289 to X.X.); the Science and Technology Planning Project of Guangdong Province, China (2023B1212060050 and 2023B1212120009 to X.X.); the Basic Research Project of Guangzhou Institutes of Biomedicine and Health, Chinese Academy of Sciences (GIBHBRP24-02 to X.X.); Guangdong Basic and Applied Basic Research Foundation (2026A1515010995 to J.W.); the Natural Science Fund of Guangdong Province (2022A1515110211 to J.W.); the Young Doctoral Starting Sail Project of the Guangzhou Municipal Science and Technology Bureau (2024A04J4197 to J.W.; 2024A04J4357 to Y.M.); and the Grants from State Key Laboratory of Respiratory Diseases (SKLRD-Z-202515 to Y.M.). We acknowledge Start-up grants from the Chinese Academy of Sciences. The funders had no role in study design, data collection and analysis, decision to publish, or preparation of the manuscript.

**Competing interests:** The authors have declared that no competing interests exist.

## Author summary

SARS-related viruses are widely found in horseshoe bats. Some can bind the human ACE2 receptor with variable affinities, whereas others bind only bat ACE2 receptors. Understanding the basis for this difference is critical for assessing spillover risk. We studied the spike protein of a bat sarbecovirus, Rc-o319, isolated from Japanese *Rhinolophus cornutus* horseshoe bats. Although Rc-o319 is genetically related to SARS-CoV-2, it is unable to bind the human ACE2 receptor. Structural analyses and functional experiments revealed that the Rc-o319 spike protein possesses a distinct receptor-binding motif (RBM). This RBM features a specialized "beta-loop" that replaces the "large-loop" found in human-infecting sarbecoviruses. The beta-loop enables high-affinity binding to its cognate *Rhinolophus cornutus* bat ACE2 receptor while simultaneously preventing optimal interaction with human ACE2. We further found that acquisition of high affinity for the human ACE2 receptor would require coordinated changes across multiple regions of the RBM, including changing the beta-loop into a large-loop structure. Together, our findings demonstrate that the Rc-o319 spike protein is highly adapted to its cognate bat ACE2 receptor and faces substantial structural constraints that limit its ability to switch to binding human ACE2.

## Introduction

The SARS-CoV-1 (also known as SARS-CoV) outbreak [1–4] and the SARS-CoV-2 pandemic have profoundly impacted global public health. Surveillance efforts have since detected multiple SARS-related coronaviruses (SARSr-CoVs) primarily in horseshoe bats (genus *Rhinolophus*) worldwide [5–9], suggesting they are the likely natural reservoir for SARSr-CoVs [10–12]. Along with SARS-CoV-1 and SARS-CoV-2, SARSr-CoVs have been classified into the *Sarbecovirus* subgenus in the *Betacoronavirus* family (https://ictv.global/taxonomy). Sarbecoviruses share considerable similarity in their spike (S) proteins with amino-acid (AA) sequence identity > 63% [13–18]. Like other coronaviruses (CoVs), sarbecovirus S-protein assembles into a large, glycosylated homo-trimer on the virion surface. Each S-protein protomer can be divided into two parts: the S1 part, responsible for cell attachment, and the S2 part, facilitating virus-cell membrane fusion [19–21]. For a sarbecovirus S-protein, its S1 part can be further divided as the N-terminal domain (NTD), receptor binding domain (RBD), Domain C (also called subdomain-1, SD-1) and Domain D (also called subdomain-2, SD-2) [17,19,22–25]. The S-RBD can be further divided into a core domain, onto which a receptor binding motif (RBM) region is attached [17,26]. The RBM primarily contacts the receptor, forming the interaction interface with the receptor [24,26–28].

For both SARS-CoV-1 [26,29] and SARS-CoV-2 [30,31], their S-RBDs bind with the human ACE2 (hACE2) receptor with high affinities in the nano-molar range, suggesting that high-affinity hACE2 binding is likely a key prerequisite for spill-over into

the human population. Of note, not all bat sarbecoviruses can bind hACE2 with high-affinity [5,7,12–16,32,33]. Phylogenetic classification of SARSr-CoVs based on S-RBD sequences has defined multiple clades [34]. Clade 1 sarbecoviruses utilize ACE2 and can be further subdivided into clade 1a (SARS-CoV-1 lineage) and clade 1b (SARS-CoV-2 lineage), both of which possess the most complete RBMs, lacking deletions of more than one amino acid. In contrast, no cellular receptor has been identified for clade 2 sarbecoviruses, which harbor deletions at two distinct regions within the RBM. Finally, clade 3 viruses contain a single loop deletion within the RBM and exhibit variable capacities for ACE2 utilization. Based on these distinct RBM deletion patterns and their associated structural features among clades 1–3 viruses, we previously classified their RBDs structurally as type-1, type-3, and type-2, respectively [17]. Similarly, the type I–IV (Roman numerals) classification of sarbecovirus RBDs proposed by Si et al. is based primarily on indel patterns within the RBM [28]. In this scheme, type I RBDs correspond to our type-1 and retain a full-length RBM; type II RBDs correspond to our type-2 and contain a single deletion in the small loop; type III RBDs harbor a deletion in the large loop; and type IV RBDs correspond to our type-3 and feature two deletions in RBM. Accordingly, Rc-o319 RBD, with a 9-amino-acid deletion in large loop, was defined by Si et al as a type III RBD [28].

Among the three types of sarbecovirus RBDs with distinct RBM structural features, four featured RBM regions have been defined, namely the large-loop (LL), lamella (LM), small-loop (SL), and finally the anchor-loop (AL) [17]. Type-1 RBD S-proteins, including those from clade 1a and 1b sarbecoviruses such as SARS-CoV-1, SARS-CoV-2 and WIV1, possess intact LL and SL regions. Type-1 RBD S-proteins often bind hACE2, albeit with varying affinities: SARS-CoV-1 and SARS-CoV-2 bind hACE2 with high affinities, whereas WIV1 binds hACE2 with a moderate affinity [35]. We found high-affinity hACE2 binding requires synergistic favourable hACE2 interactions across the four featured RBM regions. Type-2 RBD S-proteins, including those from clade 3 sarbecoviruses such as BM48–31 and BtKY72, have an intact LL but a deleted SL. They generally lack hACE2 binding but can bind selected bat ACE2 (bACE2) orthologs with intermediate affinities. Type-3 RBD S-proteins, including those from clade 2 sarbecoviruses such as HeB2013 and RmYN02, have both their LL and SL regions deleted. To date, type-3 RBD S-proteins have not been reported to bind either hACE2 or bACE2.

Ongoing surveillance in Japan has recently discovered several new SARSr-CoVs in little Japanese horseshoe bats - *Rhinolophus cornutus* (*R.cor*), based on analyses reported in 2020 using samples collected in 2013 [7]. The representative virus, Rc-o319, is distantly related to SARS-CoV-2, sharing 81.5% whole-genome identity. The nucleotide and amino acid sequences of Rc-o319 S-protein are more homologous to those of viruses belonging to the SARS-CoV-2 lineage (clade 1b) than to those of the SARS-CoV-1 lineage (clade 1a) within the *Sarbcecovirus* subgenus (**S1 Fig**). Interestingly, although its S-protein has 76.7% AA identity to SARS-CoV-2 S-protein, its RBM contains a unique nine-amino-acid deletion relative to SARS-CoV-2 that removes residues identified as critical for ACE2 binding in other SARSr-CoVs and likely alters its RBM structure [7]. Deletions in the RBMs of SARSr-CoVs have been shown to greatly affect ACE2 binding [28,36]. In this study, we surveyed binding of Rc-o319 S-protein to various ACE2 orthologs to understand its specificity. Further, we determined the structures of the Rc-o319 S-trimers and of the Rc-o319-RBD bound to *R. cornutus* ACE2 (bACE2$_{R.cor}$), thereby structurally defining the Rc-o319 RBD as a novel type of RBD. By analysing the molecular determinants of receptor specificity revealed by our structures, we identify the changes necessary to switch the specificity of the Rc-o319 S-protein to hACE2. These findings provide insights into Rc-o319's adaptations to various ACE2 orthologs and provide an assessment of its cross-species transmission potential.

## Results

### Rc-o319 S utilizes a narrow range of *Rhinolophus* bat ACE2 orthologs as receptors

We used a S:ACE2 interaction mediated cell-cell fusion assay to identify ACE2 orthologs that can interact with Rc-o319 S-protein [37]. Effector cells expressing Rc-o319 S-proteins were tested against receptor cells expressing a panel of ACE2 orthologs for fusion activity (**Fig 1A**). In this assay, several *Rhinolophus affinis* (*Ra*) and *Rhinolophus sinicus* (*Rs*) ACE2 molecules, including *Ra*5538, *Ra*9479, *Rs*1446, *Rs*3359, and *Rs*3366, were found to mediate varying levels of

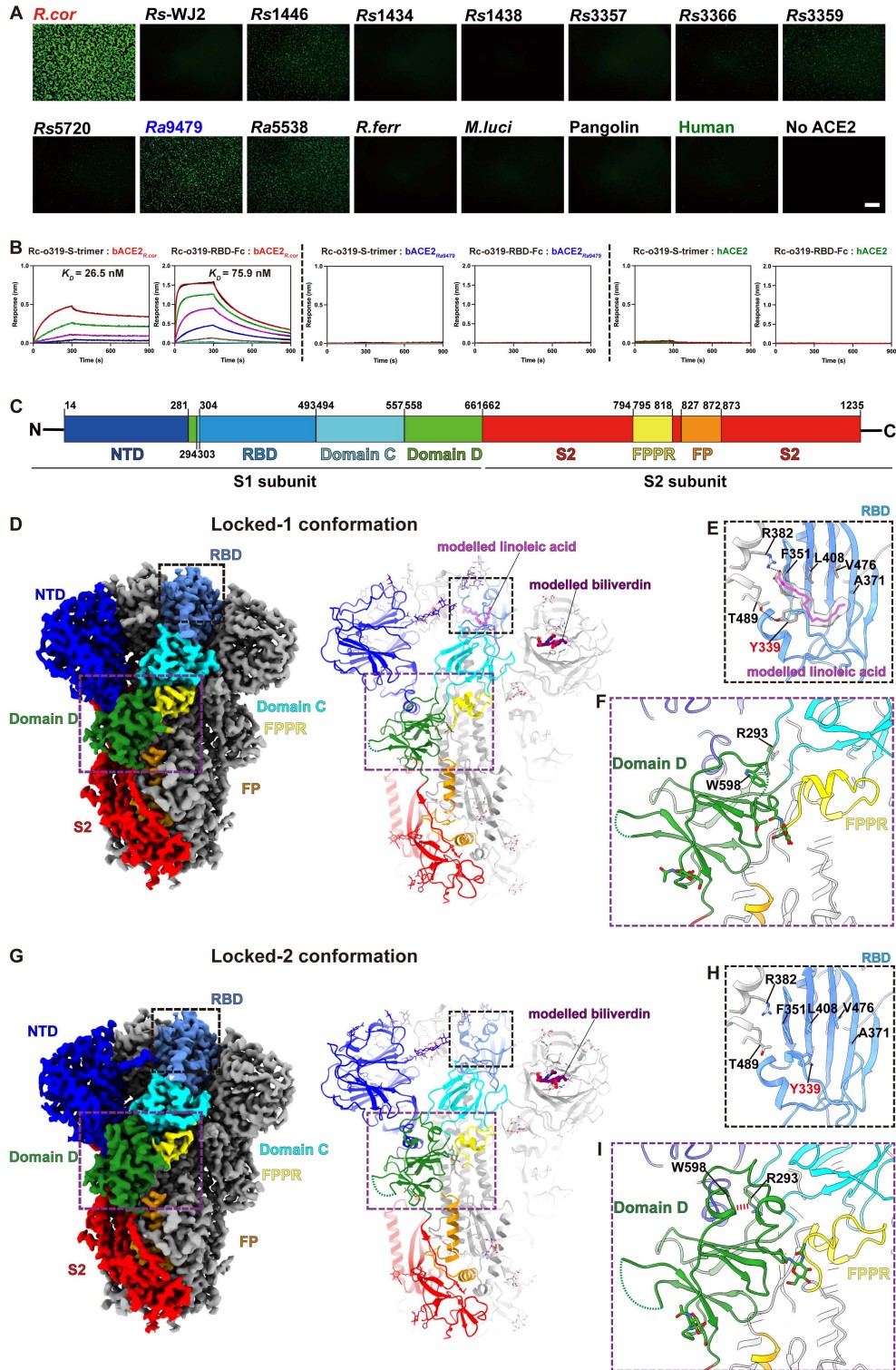

**Fig 1. Rc-o319 S-protein utilizes a narrow range of *Rhinolophus* bat ACE2 orthologs as receptors and adopts two locked, 3-RBD-down conformations. (A)** Cell-cell fusion assays using receptor cells expressing bat, pangolin or human ACE2 molecules and effector cells expressing the Rc-o319 S-proteins. Fused cells show GFP signals. Representative images of cell-cell fusion captured at 6 hours post-cell-mixing are shown. Scale bar, 500 μm. **(B)** Binding of Rc-o319 S-trimer to immobilized dimeric bACE2$_{R.cor}$, hACE2 and bACE2$_{Ra9479}$ as assessed by BLI assays. **(C)** A schematic representation showing the eight functional domains or regions of Rc-o319 S-protein. **(D)** Cryo-EM density and associated molecular model showing the

overall architecture of a Rc-o319 S-trimer in a locked-1 conformation. **(E)** Lipid binding pocket in the RBD of locked-1 Rc-o319 S-trimer. Modeled linoleic acid (magenta) interacts with the shown hydrophobic residues of the lipid binding pocket. Salt bridge and hydrogen bond are shown as dashed lines. **(F)** Structure of the Domain D region in the locked-1 Rc-o319 S-trimer. Consistent with a locked conformation, the fusion peptide proximal region (FPPR) is ordered. Residues 580-605$_{Rc-o319}$ form a large disordered loop in the locked-1 conformation. **(G)** Cryo-EM density and associated molecular model showing the overall architecture of a locked-2 Rc-o319 S-trimer. **(H)** The RBD lipid binding pocket is empty with Y339$_{Rc-o319}$ side-chain reorientated in the locked-2 conformation. **(I)** Residues 580-605$_{Rc-o319}$ refold and form an ordered structure. A stabilizing cation-π interaction (red dashed line) is formed between Domain D residue W598$_{Rc-o319}$ and Domain C-D junction residue R293$_{Rc-o319}$.

cell-cell fusion, with *Ra*9479 showing the highest activity and *Rs*3366 the lowest. However, all were substantially weaker than the cognate bACE2$_{R.cor}$ receptor. Rc-o319 S-protein is unable to utilize pangolin and human ACE2 (hACE2) orthologs for fusion. Receptor binding assays using biolayer interferometry (BLI) show that Rc-o319-RBD-Fc and Rc-o319-S-trimer bind bACE2$_{R.cor}$ with moderate-to-high affinities (**Fig 1B** left panels, 76 nM for RBD-Fc and 27 nM for S-trimer, and **S1 Table**). Although Rc-o319 appears to somewhat use bACE2$_{Ra9479}$, showing weak fusion activity in the cell-cell fusion assay. BLI assays were unable to detect bACE2$_{Ra9479}$ binding by the Rc-o319 S-trimer and dimeric Rc-o319-RBD-Fc (**Fig 1B**, middle panels). Rc-o319-S-trimer binds bACE2$_{R.cor}$ with relatively weak responses and an estimated affinity of 27 nM, tighter than that observed for the dimeric Rc-o319-RBD-Fc, indicating avidity-amplified receptor binding (**Fig 1B**, left panels). Consistent with the cell-cell fusion assay results, neither Rc-o319-RBD-Fc nor Rc-o319-S-trimer was able to bind hACE2 (**Fig 1B**, right panels). These results demonstrate that Rc-o319-S-protein has a restricted ACE2 specificity, with a clear preference towards its cognate bACE2$_{R.cor}$ receptor (**Fig 1A** and **1B**).

## Structures of the Rc-o319 S-protein identify a new type of RBM

To understand the unusual restricted receptor specificity of Rc-o319 S-protein, purified Rc-o319 S-protein ectodomain was imaged by cryo-electron microscopy (Cryo-EM) (**S2 Fig** and **S2 Table**), revealing that Rc-o319 S-trimers adopt two distinct conformations (**Figs 1D-1I**, **S2**, **S3**, **S4A**, **S4F** and **S4G**). In both conformations, the 3 RBDs in each S-trimer are all in "down" positions (**Fig 1D** and **1G**). Detailed structural analysis reveals tight trimer packing for both conformations. Based on their structural features, the two conformations differ primarily at domain D (**Figs 2A**, **2B**, **S3**, **S4**F and **S4G**) and have been identified as "locked-1" and "locked-2" conformations previously identified for both SARS-CoV-1 and SARS-CoV-2 S-trimers [22,25,38–40].

In the locked-1 Rc-o319 S-trimer, each RBD contains a density that can be modelled as a fatty acid with 18-carbon atoms, previous mass-spectrometry analysis identified it being either linoleic acid [41,42] or oleic acid [43] (**Figs 1E** and **S4C**). Consistent with previous observations, the aliphatic tail of the fatty acid is bound within the hydrophobic fatty-acid binding pocket of one RBD, while its carboxylate group forms salt-bridges with R382$_{Rc-o319}$ of a neighbouring RBD (**Figs 1E** and **S4C**); these fatty acids mediate inter-protomer interactions likely stabilizing the S-trimer in the observed locked-1 conformation. Each NTD also contains a density corresponding to a biliverdin molecule. The binding of biliverdin has been correlated with evasion of antibody immunity [44,45] (**Figs 1D**, **1G** and **S4D**). Consistent with being in a locked confor- mation, the fusion peptide proximal region (FPPR) is ordered (**Fig 1F**). Finally, consistent with the locked-1 conformation observed for the S-proteins of SARS-CoV-1 [25] and SARS-CoV-2 [40], residues 580–605$_{Rc-o319}$ form a large disordered loop within Domain D of the Rc-o319 S-trimer (**Figs 1D**, **1F** and **S4F**).

In the locked-2 conformation, no fatty acid density was observed within the RBD fatty-acid binding pocket (**Fig 1H**). This is accompanied with a restructuring of the fatty-acid binding pocket (**Figs 1H** and **S4C**) as observed when the fatty- acid binding pockets were vacated in other sarbecovirus S-trimers [41,46]. Each NTD still contains a density modelled as a biliverdin (**Fig 1D** and **1G**). The FPPR is ordered (**Fig 1I**). Differently, the disordered Domain D loop in the locked-1 con- formation refolds and forms a large ordered helical structure, packing against the core of the Domain D and the junction between Domains C and D with non-interupted cryo-EM density (**Figs 1F**, **1I** and **S4G**) [25,40].

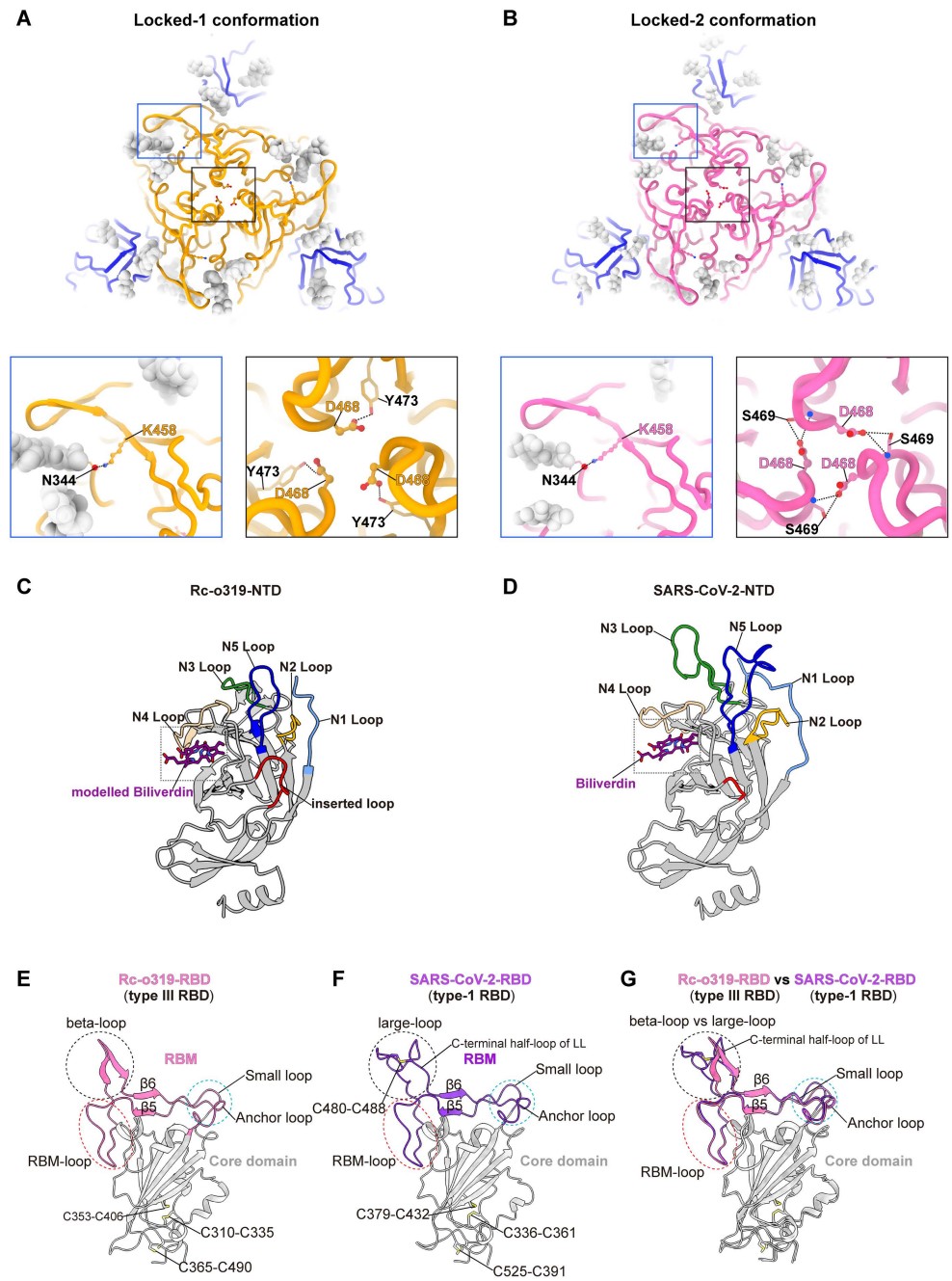

**Fig 2. Rc-o319 S-trimer structures reveal an RBD featuring a novel RBM and an NTD with altered loops. (A-B)** Top-views of locked-1 and locked-2 Rc-o319 S-trimer structures. N-linked glycans are shown as light gray spheres. NTDs are colored blue, RBDs are colored orange (locked-1, **A**) or pink (locked-2, **B**). The lower panels show intra- or inter-protomer interactions involving RBD residues. $K458_{Rc\text{-}o319}$ forms an inter-protomer hydrogen bond with $N344_{Rc\text{-}o319}$ of an RBD of a neighboring protomer in both locked-1 and locked-2 S-trimers. In the locked-1 conformation, $D468_{Rc\text{-}o319}$ forms an intra-protomer hydrogen bond with $Y473_{Rc\text{-}o319}$; in the locked-2 conformation, $D468_{Rc\text{-}o319}$ instead mediates two inter-protomer hydrogen bonds with $S469_{Rc\text{-}o319}$ of an adjacent protomer. **(C-D)** Comparison of Rc-o319 **(C)** and SARS-CoV-2 **(D)** N-terminal domain (NTD) structures. The five highly variable NTD loops (N1-N5) are shown in the Rc-o319 and SARS-CoV-2 (PDB: 7XU4) NTD structures. A four-amino-acid residue insertion is highlighted in the Rc-o319 NTD in red. **(E)** The structure of the Rc-o319 RBD. The receptor binding motif (RBM) of the RBD is colored pink and the core of the RBD is colored gray. Disulfide bonds within the RBD are indicated. Featured structural elements, beta-Loop (BL), small loop (SL), Lamella (LM) sheets, β5 and β6, and the large RBM loop are indicated. **(F-G)** For comparison, the SARS-CoV-2 RBD (PDB: 7XU4) is shown **(F)** and superposed on the Rc-o319 RBD structure **(G)**.

A structural change between locked-1 and locked-2 conformations in the Domain C-D junction region can be observed similar to previous observations on the S-proteins of SARS-CoV-1 and SARS-CoV-2 [25,40] (**Figs 4F, 4G, and S4A**). Near the 3-fold axis, to the top of the S-trimer, $D468_{Rc-o319}$ forms an intra-protomer hydrogen bond with $Y473_{Rc-o319}$ in the locked-1 conformation (**Fig 2A**). Whereas in locked-2 conformation, $D468_{Rc-o319}$ mediates two inter-protomer hydrogen bonds with $S469_{Rc-o319}$ of a neighbouring S-protomer (**Fig 2B**). Of Note, in both conformations, $K458_{Rc-o319}$, equivalent to $Q493_{SARS2}$, previously identified as an important determinant for ACE2 specificity [17,47,48], is found to form an inter-protomer hydrogen bond to the glycan attaching to $N344_{Rc-o319}$ of a neighbouring S-protomer (**Fig 2A** and **2B, the second panels**). Inter-protomer hydrogen bonds likely contribute to S-trimer stability.

In both Rc-o319 S-trimer conformations, their NTDs are well resolved, revealing five loops that are markedly shorter than those in SARS-CoV-2 (**Figs 2C**, **2D** and **S4E**). In addition, a four-amino-acid insertion (residues $189–192_{Rc-o319}$) is identified in the Rc-o319 NTD, forming an "inserted loop" exposed to the exterior of the trimer (**S4B Fig**). It's well-established that the SARS-CoV-2 S-protein NTD is under strong immune pressure [49,50]. The truncated and elongated loops observed in the Rc-o319 S-protein NTD may therefore confer a distinct antigenic profile, although whether these features reflect immune-driven selection in bats remains to be determined.

Finally, Rc-o319 S-trimer structures show that their RBDs form structures distinct from previous characterized sarbecovirus RBDs (**Figs 2E**-**2G** and **S5**). In contrast to type-3 RBDs, where 13- to 14-amino-acid deletions relative to SARS-CoV-2 eliminate the large-loop (LL) (**S5C** and **S5F**-**S5G Fig**), the Rc-o319 RBD retains an extension in a similar region as the LL of Type-1 RBDs, despite a nine-amino-acid deletion (**Fig 2E and 2G**). In the type-1 RBD, the LL is formed by two half-loops stabilized by an intra-loop disulfide bond (**Fig 2F**). By contrast, the equivalent extension in the Rc-o319 RBD comprises two small antiparallel β-sheets connected by a short loop (**Figs 2E** and **3**). We designate this distinct extension as a beta-loop (BL) structure. Due to the nine-amino-acid deletion, the stabilizing disulfide bond found in the LL is lost. The shorter BL occupies a position roughly corresponding to the C-terminal half-loop of the LL in type-1 RBDs (**Figs 2E**-**2G** and **3D**), albeit with notable conformational and tilting angle differences (**Figs 2G** and **3D**). This distinct beta-loop (BL) structure, by comparison with the LL structure in type-1 RBDs (**Fig 2F**), suggests that the Rc-o319-RBD likely binds ACE2 with distinct interactions.

## Structure of the Rc-o319-RBD:bACE2$_{R.cor}$ complex

To further understand receptor binding by the Rc-o319 S-protein, we determined a structure of Rc-o319-RBD in complex with its cognate bACE2$_{R.cor}$ receptor (**Figs 3** and **S2B**). This structure uncovers several previously unrecognized interactions at the Rc-o319-RBD:bACE2$_{R.cor}$ interface, even though the molecular surfaces employed for binding largely mirror those in other sarbecovirus RBD:ACE2 complexes (Fig 3A and 3B). These interactions are mediated by the specialized RBD BL and RBM-loop and a glycan linked to Asn38 of bACE2$_{R.cor}$ (Fig 3A, 3E and Fig 3H-3I) (interfacial residues in the RBD and ACE2 are denoted using single-letter and three-letter amino acid codes, respectively). The buried surface area (BSA) of the complex was calculated as 1005 $Å^2$, which includes a contribution of 252 $Å^2$ from the binding of the Asn38-linked glycan. In line with previous interface analysis, the binding interface can be divided into four regions, namely the beta-loop (BL), lamella (LM), anchor loop (AL) and small loop (SL) regions (Fig 3F-3I).

The beta-loop (BL) of Rc-o319 RBD engages highly distinctive interactions (**Fig 3A** and **3F**). Due to its distinct structure and protrusion from the RBM, the BL contacts the bACE2$_{R.cor}$ near Lys27, forming a hydrogen bond with the side chain of Lys27 (**Fig 3A** and **3F**). In contrast, LL in type-1 RBDs contacts ACE2 near residue 24, more towards the ACE2 N-terminus (**Fig 3J**). Distinctively, BL residue $Y447_{Rc-o319}$ forms a charged hydrogen bond with Asp31 of bACE2$_{R.cor}$ (**Fig 3A** and **3F**). Of note, the positively charged residue $K454_{Rc-o319}$, unique to the Rc-o319 RBM, extends from the BL to form a salt-bridge with $Glu75_{Rc-o319}$ of bACE2$_{R.cor}$ (**Fig 3A** and **3F**) and is positioned in close proximity to the negatively charged bACE2$_{R.cor}$ residues Asp31 and Glu35 (**Fig 3F** and **3G**). Changing this residue ($K454Y_{Rc-o319}$) to the corresponding residue Y in SARS-CoV-2 substantially reduced bACE2$_{R.cor}$ binding (**S9B Fig and S4 Table**).

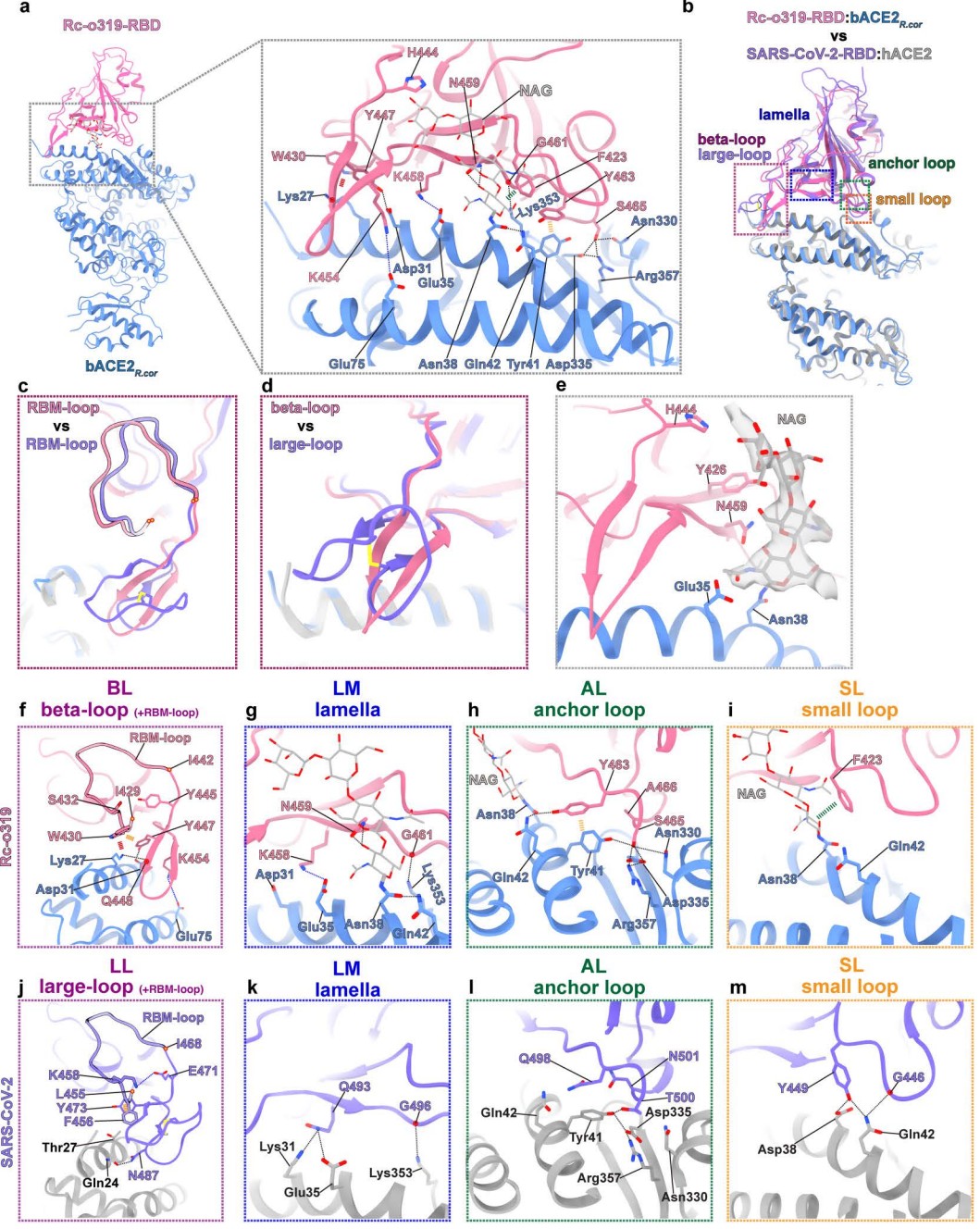

**Fig 3. Cryo-EM structure of the Rc-o319-RBD:bACE2$_{R.cor}$ complex with a detailed comparison of RBD:ACE2 interactions versus those in the SARS-CoV-2-RBD:hACE2 complex.** (A) Molecular structure of a monomeric Rc-o319-RBD:bACE2$_{R.cor}$ complex extracted from the dimeric Rc-o319-RBD:bACE2$_{R.cor}$ complex structure resolved by cryo-EM (see S2 Fig). Detailed interactions formed between the Rc-o319-RBD and bACE2$_{R.cor}$ are shown with hydrogen bonds and salt bridges shown as dashed lines. **(B)** Superposition of the Rc-o319-RBD:bACE2$_{R.cor}$ and SARS-CoV-2-RBD:hACE2 complex structures. The interface areas between the RBD and ACE2 in the two structures are divided into four regions: beta-loop (BL in Rc-o319-RBD)/large-loop (LL in SARS-CoV-2-RBD) (purple box), lamella (LM, blue box), anchor-loop (AL, green box), and small-loop (SL, orange box). **(C-D)** Comparison of the Rc-o319 RBM-loop and beta-loop (BL) with the RBM-loop and large-loop (LL) of SARS-CoV-2. **(E)** An N-linked glycan (shown in grey stick representation overlayed with white cryo-EM density) attached to Asn38 of bACE2$_{R.cor}$ is involved in the binding interface with Rc-o319-RBD, representing a unique structural feature. **(F-I)** Detailed interactions identified in the beta-loop+RBM-loop (BL, purple box), lamella (LM, blue box), anchor-loop (AL, green box), and small-loop (SL, orange box) regions of the interface between Rc-o319-RBD and bACE2$_{R.cor}$. **(J-M)** For comparison,

detailed interactions are shown for the corresponding regions in the interface between SARS-CoV-2-RBD and hACE2. Hydrogen bonds, pi-pi (sugar-pi) interactions and cation-pi interactions are indicated by black, yellow (green) and red dashed lines, respectively.

Distinct from SARS-CoV-2, the large Rc-o319 RBM-loop (**Figs 2E**, **3C**, **3F** and **S7**) adjacent and N-terminal to the BL contributes to bACE2$_{R.cor}$ binding and contacts the BL structure with specialized interactions (compare **Fig 3F** and **3J**): W430$_{Rc-o319}$ in the RBM-loop forms a cation-π interaction with Lys27 of bACE2$_{R.cor}$ and likely stabilizes the BL via a π-π interaction with the BL residue Y447$_{Rc-o319}$ (**Figs 3F** and **S7**B and **see below**).

In the lamella (LM) region, K458$_{Rc-o319}$ forms a salt-bridge with bACE2$_{R.cor}$ Glu35 (**Fig 3G**). Distinctively, in this region, the Asn38-glycan is part of the ACE2-RBD interface, forming hydrogen bonds with N459$_{Rc-o319}$ (**Fig 3G**). Also in this region, Lys353$_{Rc-o319}$ forms a hydrogen bond with the G461$_{Rc-o319}$ backbone (**Fig 3G**). In the anchor-loop (AL) region, a hydrogen bond network is observed among bACE2$_{R.cor}$ residues Tyr41, Asn330, Glu335, Arg357 and the RBD residue S465$_{Rc-o319}$ (**Fig 3H**). Further, the glycan attached to Asn38 of bACE2$_{R.cor}$ forms an inter-molecular hydrogen bond with bACE2$_{R.cor}$ residue Gln42 and also hydrogen bonds with RBD residue Y463$_{Rc-o319}$ (**Fig 3H**). Finally, in the small-loop (SL) region, F423$_{Rc-o319}$ forms a sugar-π contact with the glycan attached to Asn38 (**Fig 3I**). These interactions, by comparison with those used by SARS-CoV-2 to engage ACE2 (Fig 3J-**3M**), show that Rc-o319 RBD engages its cognate bACE2$_{R.cor}$ receptor with unique characteristics.

## Molecular determinates underlying Rc-o319 specificity for bACE2$_{R.cor}$

To understand the specificity of the Rc-o319 S-protein towards the bACE2$_{R.cor}$ receptor. We employed a mutagenesis approach to systematically change RBM regions to match those of SARS-CoV-2 (BL change = swapping of 443–456$_{Rc-o319}$ "AHYDYQVGTQFKSS" to 469–491$_{SARS2}$ "STEIYQAGSTPCNGVEGFNCYFP"; SL change = F423Y$_{Rc-o319}$; LM change = K458Q$_{Rc-o319}$; AL change = S465T$_{Rc-o319}$ + A466N$_{Rc-o319}$ + H470Y$_{Rc-o319}$; RBM-loop change = W430F$_{Rc-o319}$ + S432K$_{Rc-o319}$). These variants were used to assess hACE2 and bACE2$_{R.cor}$ binding by BLI assays using RBD-Fc proteins, as well as receptor utilization by cell-cell fusion and pseudovirus entry assays using full-length S-proteins (**Figs 4M** and **S13**–**S15**). A similar approach has been previously used to elucidate how high-affinity hACE2 binding may be acquired by S-proteins with type-2 RBDs, as we defined previously [17], from clade 3 sarbecoviruses.

We found that changing AL and SL regions individually had very little effect in improving hACE2 binding (**Fig 4D** and **4E, the second panels, and** **S3 Table**). The SL change retained very strong bACE2$_{R.cor}$ utilization (**Fig 4E, the third** and **fifth panels**), whereas the AL change reduced bACE2$_{R.cor}$ utilization somewhat in cell-cell fusion (**Fig 4D, the third** and **fifth panels)** and pseudovirus entry assays (Fig 4M). Interestingly, the AL change very slightly increased bACE2$_{R.cor}$ binding when assayed using both Rc-o319-RBD-Fc and the Rc-o319 S-trimer (**Figs 4D, the first panels, S10C and** **S14F, and** **S3**, **S5** and **S9 Tables**). The discrepancy between the increased BLI affinity and the reduced cellular activity may reflect differences in assay conditions. In the cellular context, phenotypes may also be influenced by S-protein-related factors beyond receptor binding, such as S-protein stability, efficiency of surface expression, incorporation into pseudovirus particles, and activation by cellular proteases, which could contribute to the observed discrepancy (see the Limitations section). Changing the LM region alone (the LM change) reduced bACE2$_{R.cor}$ binding and utilization (**Fig 4C, the first, third** and **fifth panels)** but still failed to confer hACE2-mediated fusion activity and pseudovirus entry (**Fig 4C, the fourth** and **fifth panels**).

Swapping of the Rc-o319 BL region to the LL sequence of SARS-CoV-2-RBD (the BL change), appears to confer Rc-o319 with the ability to utilize hACE2 for cell fusion activity at the expense of substantially reduced fusion activity mediated by its cognate bACE2$_{R.cor}$ receptor (**Fig 4B**). Consistently, the BL change conferred hACE2 mediated pseudovirus entry, while pseudovirus entry mediated by the cognate bACE2$_{R.cor}$ reduced by ~52-fold (**Fig 4M**). Despite conferring detectable cell-cell fusion activity and pseudovirus entry, hACE2 binding by the dimeric BL variant of the Rc-o319 RBD-Fc

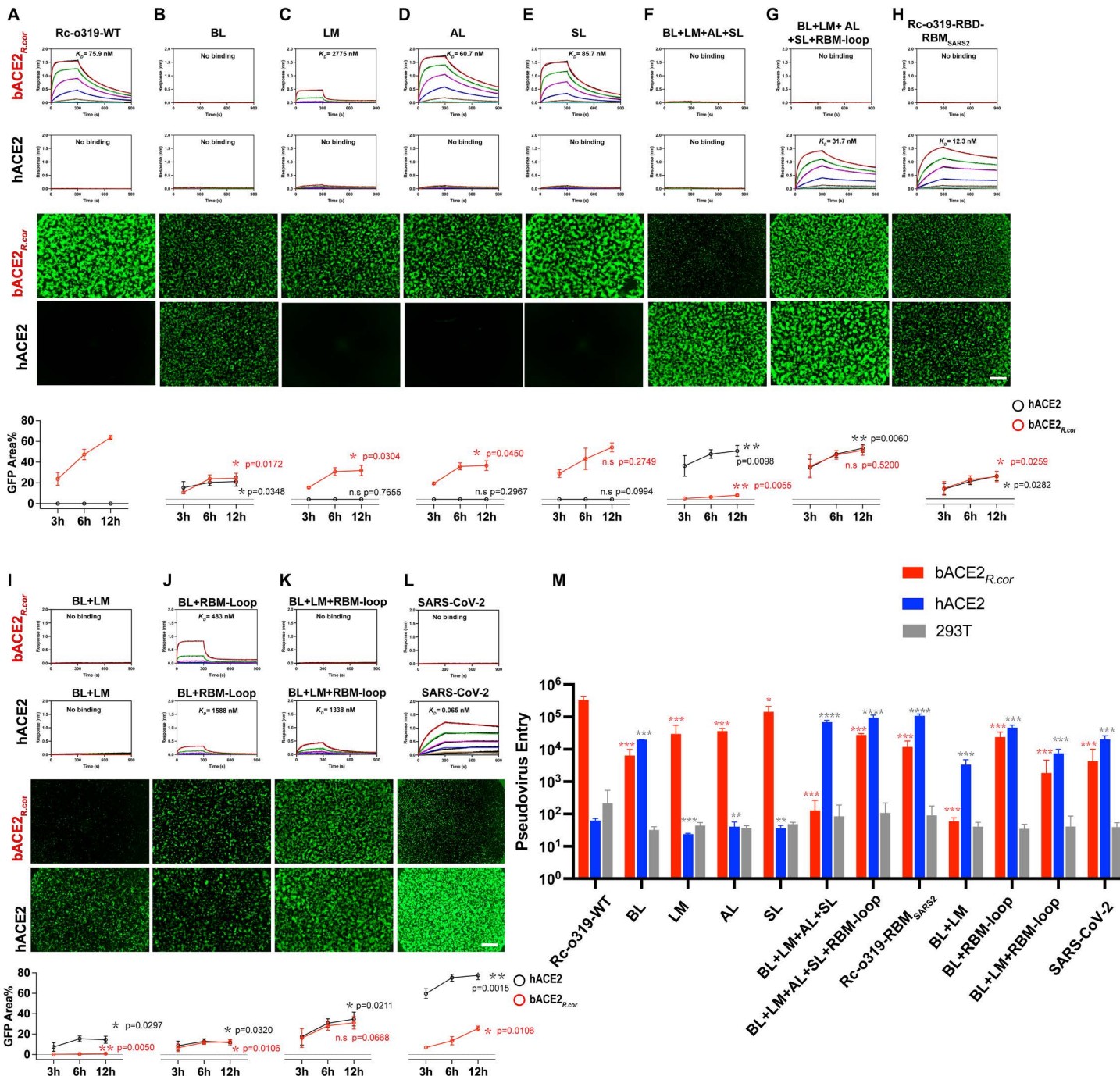

**Fig 4. ACE2 utilization by wild-type and variant Rc-o319 RBD-Fc in BLI and wild-type and variant Rc-o319 S-proteins in cell-cell fusion and Pseudovirus entry assays. (A-L)** Top two panels, binding of bACE2$_{R.cor}$ and hACE2 by wild-type (WT) and variant Rc-o319-RBD-Fc proteins as assessed by BLI assays. Dimeric Rc-o319-RBD-Fc proteins were immobilized on the BLI sensors and tested binding against dimeric *Rhinolophus cornutus* bat ACE2 (bACE2$_{R.cor}$) or hACE2 protein as the analyte in solution. bACE2$_{R.cor}$ and hACE2 were three-fold serial diluted from 3000 to 4.1 nM. $K_D$ values estimated from each binding experiment are shown alongside the corresponding binding curves. Lower two panels, Representative images of cell-cell fusion captured at 12 hours post-cell-mixing are shown. Scale bar, 500 µm. Effector cells expressing WT or variant Rc-o319 S-proteins were tested against receptor cells expressing either bACE2$_{R.cor}$ or hACE2. The bottom panels show the quantification of cell-cell fusion measuring the GFP+ area at 3, 6, 12 hours (h) post-cell-mixing. Data are presented as mean ± s.d., n = 2. The *P* values were determined by unpaired t-test comparing wild type (WT) with each Rc-o319 variant. *$P < 0.05$, **$P < 0.01$, ***$P < 0.001$; NS, not significant. **(M)** Pseudovirus assay. Entry of VSV particles pseudotyped

with Rc-o319 S-protein variants into HEK293T cells transiently expressing hACE2 or bACE2$_{R.cor}$ and co-expressing TMPRSS2. Luciferase activity was measured 24 h post-transduction. Data are presented as mean ± s.d. (n = 3). *P* values were calculated using unpaired t tests comparing wild-type (WT) with each Rc-o319 variant. *P < 0.05, **P < 0.01, ***P < 0.001 and ****P < 0.0001.

protein remained below the detection limit of the BLI assay, while bACE2$_{R.cor}$ binding was markedly reduced and fell below the detection limit of the BLI assay (**Fig 4B, the first panels**). A subsequent BLI assay using the BL variant of the Rc-o319 S-trimer detected weak hACE2 binding (**S10B Fig** and **S5 Table**), likely made detectable by amplification through trivalent binding-mediated avidity, underscoring a dependence of assay sensitivity on binding valency. As cell-based assays involve highly multivalent interactions mediated by multiple S-trimers and receptors, avidity-enhanced binding renders them substantially more sensitive to weak interactions (also see the Limitations section).

Therefore, the above results show that replacing each RBM region singly with its SARS-CoV-2 counterpart in the Rc-o319 RBD-Fc failed to confer high-affinity binding to hACE2. This finding is consistent with our previous observations that high-affinity hACE2 binding requires a sarbecovirus RBD to establish optimal contacts across its all four RBM regions, including LL/BL, LM, AL, and SL [17]. A similar requirement for synergistic interactions across multiple RBM regions to achieve high-affinity receptor binding has also been observed in merbecoviruses [51].

However, unexpectedly, the BL + LM + AL + SL variant, designed to introduce optimal interactions across all four RBM regions, exhibited enhanced hACE2 utilization in cell-cell fusion and pseudovirus entry assays, yet its hACE2 binding remained too weak to be detected by BLI (**Fig 4F, the second** and **fourth panels**). The BL + LM + AL + SL variant of Rc-o319-RBD-Fc protein also showed no bACE2$_{R.cor}$ binding by BLI, and consequently exhibited very little bACE2$_{R.cor}$ mediated fusion activity (**Fig 4F, the first** and **third panels**). Therefore, in the case of Rc-o319 S-protein, determinants outside of the BL, LM, AL, and SL regions likely hinder high-affinity hACE2 binding.

Structural comparison shows that W430$_{Rc-o319}$ and S432$_{Rc-o319}$ in the large RBM-loop [amino acid (AA) 429–442$_{Rc-o319}$, corresponding to AA 455–468$_{SARS2}$], located N-terminal to the BL, are unique to the Rc-o319 RBD (**Figs 3C**, **3F**, **S5**G and **S7**). In SARS-CoV-2-RBD, the corresponding residues F456$_{SARS2}$ and K458$_{SARS2}$, respectively interact with the LL residues Y473$_{SARS2}$ and E471$_{SARS2}$ (**Figs 3J**, **S7**A and **S7D**). These interactions likely stabilize the SARS-CoV-2 LL structure. We introduced these residues into the Rc-o319 BL + LM + AL + SL variant as the W430F$_{Rc-o319}$ + S432K$_{Rc-o319}$ change (collectively named as the "RBM-loop" change). BLI binding results show that the dimeric BL + SL + LM + AL + RBM-loop variant of Rc-o319-RBD-Fc is able to bind hACE2 with a much increased affinity of 31.7 nM (**Fig 4G, the second panel**). We also attempted the swap of Rc-o319 RBM for the SARS-CoV-2 RBM (Rc-o319-RBD-RBM$_{SARS2}$), as expected, the Rc-o319-RBD-Fc variant with the swapped RBM also showed high-affinity binding to hACE2 with only slightly higher affinity (12.3 nM) than the BL + LM + AL + SL + RBM-loop variant (**compare Fig 4G** and **4H, the second panels**). Consistently, both BL + LM + AL + SL + RBM-loop and RBM swapped Rc-o319-RBD-RBM$_{SARS2}$ show effective hACE2 mediated cell-fusion activities (**Fig 4G** and **4H, the fourth** and **fifth panels**). Therefore, our mutagenesis results identify residues in the RBM-loop of Rc-o319, N-terminal to the BL, that are incompatible with the LL structure of type-1 RBDs, thereby preventing the LL from mediating optimal ACE2 interactions (**S7C, S7D** and **S8C Figs**).

To further clarify the role of the RBM-loop in hACE2 recognition, we introduced the RBM-loop change alone into the Rc-o319 S-protein. This change did not confer hACE2-dependent cell-cell fusion (**S8 Fig**), consistent with structural analyses indicating that the RBM-loop does not make direct contacts with ACE2 (**Fig 3**). No detectable hACE2 binding was observed for the Rc-o319 BL-variant RBD-Fc (**Fig 4B**), and a similar lack of hACE2 binding was also observed for the Rc-o319 BL + LM-variant RBD-Fc in BLI assays (**Fig 4I**), despite the corresponding S-proteins mediating clear hACE2-dependent cell-cell fusion and pseudovirus entry (**Fig 4B**, **4I** and **4M**). When the RBM-loop change was introduced into the Rc-o319 BL- or BL + LM-variant RBD-Fc (i.e., BL + RBM-loop and BL + LM + RBM-loop variants), weak but detectable hACE2 binding could be detected in BLI assays (**Fig 4B** and **4I**–**4K**). These results confirm that the RBM-loop plays a modulatory role, essential for mediating high-affintiy ACE2 interaction.

## Utilization of bACE2$_{R.cor}$ by different types of sarbecoviruses

Our structural and biochemical analyses show that the Rc-o319 S-protein has a highly restrictive specificity towards its cognate bACE2$_{R.cor}$ receptor, we further characterized the potential of bACE2$_{R.cor}$ to support diverse sarbecoviruses (Fig 5). A panel of sarbecovirus S-proteins was surface-expressed on effector cells and tested for cell-cell fusion activity against receptor cells expressing bACE2$_{R.cor}$ (**Fig 5A**). The panel included S-proteins with type-1 RBDs [17] from clade 1a and 1b sarbeviruses, including SARS-CoV-1, LYRa11, SHC014, WIV1, RaTG13, BANAL-20-52, BANAL-20-236, and the SARS-CoV-2 ancestral strain; S-proteins with type-2 RBDs [17] from clade 3 sarbeviruses, including BtKY72, BM48-31, RfGB02, and RhGB07; and S-proteins with type-3 RBDs [17] from clade 2 sarbeviruses, including GX2013, HeB2013, YN2013, and RmYN02. The results show that bACE2$_{R.cor}$ supported receptor binding and mediated cell fusion activities of several clade 1 sarbecovirus S-proteins with type-1 RBDs (**Fig 5B** and **5C**). However, potent fusion activity was only observed for several S-proteins of viruses from the SARS-CoV-2 lineage (clade 1b). Although the SARS-CoV-1 S-protein exhibits only intermediate bACE2$_{R.cor}$-mediated cell-cell fusion activity, it is nevertheless the most potent among S-proteins from viruses within the SARS-CoV-1 lineage (clade 1a) (Fig 5A-**5C**). Among bACE2$_{R.cor}$ utilizing S-proteins, only the BANAL-20-236

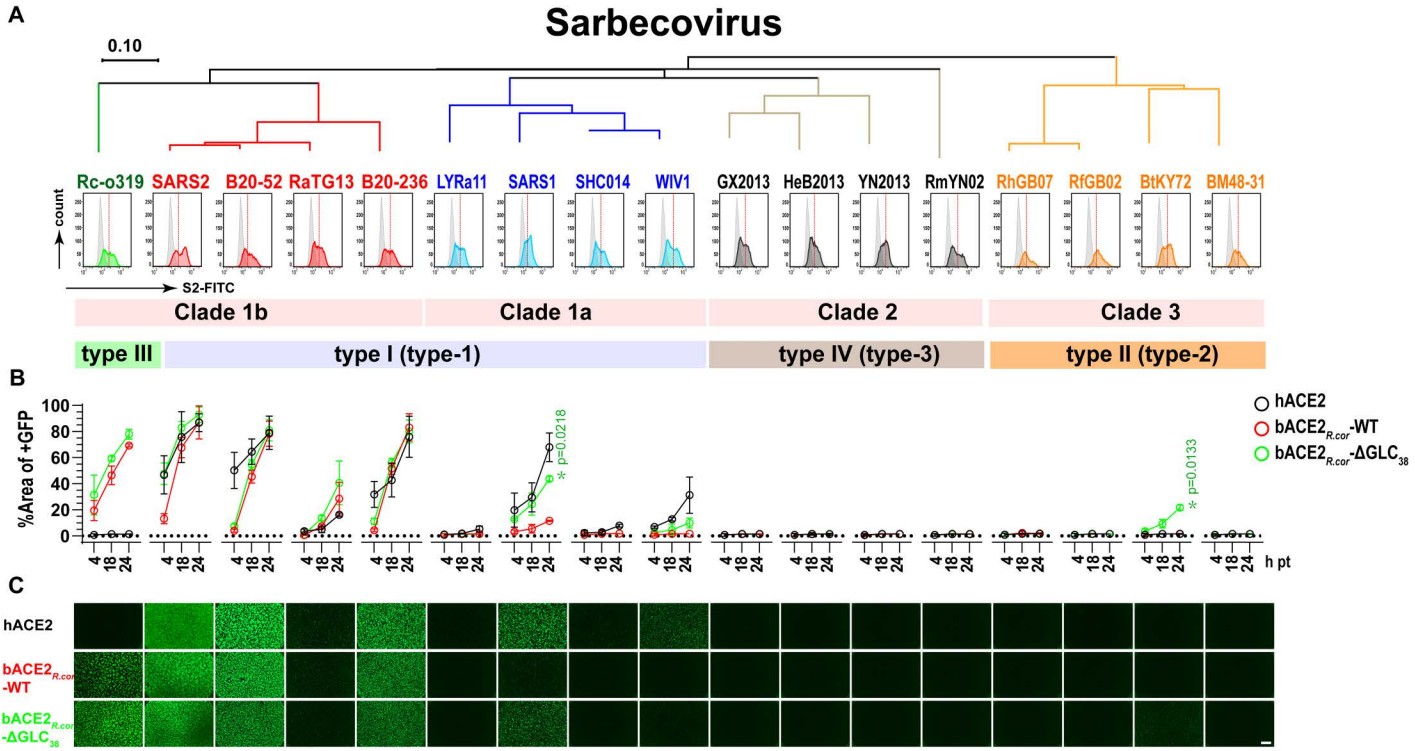

**Fig 5. bACE2$_{R.cor}$ utilization by different sarbecoviruses S-proteins. (A)** Different sarbecovirus S-proteins are sorted based on S-protein phylogenetics (clade 1—3) and grouped by RBD types (I-IV, as defined by Si et al [28]. and 1–3, as defined by Wang et al [17].). Type I (type-1) RBD strains are shown in red (SARS-CoV-2 clade, Clade 1b) and blue (SARS-CoV-1 clade, clade 1a), type II (type-2) strains in orange, type IV (type-3) strains in brown and type III strains in green. Quantification of S-protein expression on the cell surface of effector cells by flow cytometry using an S2 antibody as the probe. Flow cytometry was repeated three times with representative data shown. **(B)** Effector cells shown in (A) were tested against receptor cells expressing hACE2 (gray line), bACE2$_{R.cor}$ (red line) and bACE2$_{R.cor}$- ΔGLC$_{38}$ (green line, bACE2$_{R.cor}$ with the Thr40Ala mutation to remove the Asn38-glycan). Cell-cell fusion was quantified by assessing GFP+ areas (y-axis) at 4, 18, and 24 hours (h, x-axis) post-cell-mixing (pt). Data are presented as mean ± s.d., n = 2. P values were calculated using unpaired t-tests comparing fusion activities mediated by bACE2$_{R.cor}$ and bACE2$_{R.cor}$-ΔGLC$_{38}$. *$P < 0.05$, **$P < 0.01$, ***$P < 0.001$; NS, not significant. (C) Representative images of cell-cell fusion captured at 18 hours post-cell-mixing are shown for receptor cells expressing different ACE2. Scale bar, 500 μm.

S-trimer was found to bind bACE2$_{R.cor}$ in BLI assays, whereas S-trimers from BANAL-20-52, SARS-CoV-2, and SARS-CoV-1 exhibited no detectable binding to bACE2$_{R.cor}$ despite displaying robust cell-cell fusion activity (**Figs 5C**, S11B, S11C, S11E **and** S11F**, and** S6 Table). These observations indicate that, in addition to receptor binding, complex cell-associated factors, potentially including S-protein expression level, protein stability, and susceptibility to activation by cellular proteases, can critically influence S-protein-mediated membrane fusion (also see the Limitations section).

The Asn38-linked glycan in bACE2$_{R.cor}$ is involved in S-protein binding, a feature observed for the first time between an ACE2 and a sarbecovirus S-protein. To understand the effect of this glycan on S-protein function, we removed the glycan by introducing a Thr40Ala change (designated as 'bACE2$_{R.cor}$-ΔGLC$_{38}$'). In cell-cell fusion assays, the bACE2$_{R.cor}$-ΔGLC$_{38}$ variant receptor showed a tendency to mediate enhanced fusion activities for all the sarbecovirus S-proteins capable of utilizing the WT bACE2$_{R.cor}$ (**Fig 5B and 5C**). BLI assay results show that the removal of the Asn38-glycan was able to enhance bACE2$_{R.cor}$ binding not only for Rc-o319 but also for the BANAL-20-52 and BANAL-20-236 S-trimers (S11A, **11C and 11E Fig and S6 Table**). In particular, although unable to utilize WT bACE2$_{R.cor}$, BtKY72 S-protein, with a type-2 RBM, of a clade 3 sarbevirus, was able to utilize bACE2$_{R.cor}$-ΔGLC$_{38}$, albeit exhibiting only very weak cell fusion activity. Consistently, BLI binding results show that the BtKY72 S-trimer was unable to bind bACE2$_{R.cor}$, but gained bACE2$_{R.cor}$-ΔGLC$_{38}$ binding with an estimated affinity of 11.8 nM (**S11H Fig, the second panel, and S6 Table**).

To further assess the functional role of the Asn38-glycan, an N-linked glycosylation site was introduced into a *Rhinolophus affinis* bat ACE2 (bACE2$_{Ra9479}$), by substituting Asp38 with Asn, generating bACE2$_{Ra9479}$-GLC$_{38}$. In both cell-cell fusion and BLI assays, BtKY72 and SARS-CoV-1 S-proteins exhibited substantially reduced fusion activity utilizing bACE2$_{Ra9479}$-GLC$_{38}$ compared with the WT bACE2$_{Ra9479}$ receptor (**S16B**, **S16**F; **S11**J and **S11K Figs, and S7 Table**). In parallel, we swapped the RBMs of SARS-CoV-1 and BtKY72 S-proteins with that of Rc-o319, generating chimeric SARS1$_{Rc-o319RBM}$ and BtKY72$_{Rc-o319RBM}$ S-proteins. Although being expressed by the cell (**S16**I and **S16J Fig**), the RBM swapping appeared to render the SARS1$_{Rc-o319RBM}$ S-protein non-functional, unable to mediate cell-cell fusion mediated by either bACE2$_{Ra9479}$ or bACE2$_{R.cor}$ receptor and their Asn38-glycan variants (**S16**C and **S16D Fig**). The BtKY72$_{Rc-o319RBM}$ S-protein showed substantially reduced utilization of the bACE2$_{Ra9479}$ and bACE2$_{Ra9479}$-GLC$_{38}$ receptors (**S16F and S16H Fig**). In contrast, its utilization of both bACE2$_{R.cor}$ and bACE2$_{R.cor}$-ΔGLC$_{38}$ was only marginally enhanced compared with the WT BtKY72 S protein, yet remained substantially weaker than that of the Rc-o319 S protein (**Figs 5C, S16E and S16G**). These results highlight compatibility constraints between different RBMs and S-protein backbones, indicating that RBM changes can adversely affect overall S-protein function.

## Discussion

In this study, our receptor binding and structural analyses show that Rc-o319 S-protein exhibits a clear preference for its cognate bACE2$_{R.cor}$ receptor with marked reduced utilization of *R. sinicus* and *R. affinis* ACE2 receptors. Rc-o319 S-protein showed no utilization of human and pangolin ACE2s. A closer relationship among Asian *Rhinolophus* bats [52] probably allowed Rc-o319 S-protein to cross-bind *R. sinicus* and *R. affinis* ACE2s (see **S17 Fig** and **S10 Table**). Previously, it has been shown that weak binding of SARS-CoV-2 S-protein to mouse ACE2 (mACE2) was unable to facilitate effective mouse infection, and increased mACE2 binding, thorugh the N501Y$_{SARS2}$ change, was key to effective mouse infection [53]. Therefore, weak *R. sinicus* and *R. affinis* ACE2 binding may not allow effective infection of *R. sinicus* and *R. affinis* bats by the Rc-o319 virus.

Our structural analyses reveal that the Rc-o319 S-trimer adopts two distinct conformations: a locked-1 conformation with fatty-acid-bound RBDs and a locked-2 conformation in which those fatty-acid binding pockets are unoccupied. Previous cryo-EM studies of bat sarbecovirus S-trimers have nearly always captured them in locked-2 conformations with and without fatty acid bound [17]. Locked-1 conformation has only been reported for the bat sarbecovirus WIV1 S-trimer [54]. Both locked-1 and locked-2 conformations have been observed for SARS-CoV-1 and SARS-CoV-2 S-trimers. The locked-1 conformation appears to be always identified with fatty acid bound, whereas the locked-2 conformation has been

PLOS Pathogens

resolved with or without fatty acid bound. Interestingly, these two conformations appear to be more difficult to capture for SARS-CoV-1 [25] and SARS-CoV-2 [22,40,55] S-trimers and their capture often requires deliberate stabilization of the S-trimers. The predominance of locked conformations in bat sarbecovirus S-trimers, versus them being rare conformations in human-infecting sarbecoviruses, has been proposed to reflect likely differences in S-trimer stability [17]. We have also proposed that tightly packed S-trimers in locked conformations facilitate viral egress [40]. As the preferred conformation for bat sarbecovirus S-proteins, these locked conformations have also been postulated to represent adaptations for fecal-oral transmission and immune evasion in their natural hosts [54,56].

Due to residue-31 of bat ACE2 molecules often being a negative charged Asp or Glu, including in bACE2$_{R.cor}$, a positively charged LM region, such as K458$_{Rc-o319}$ (corresponding to Q493$_{SARS2}$), has been noted as an important determinant for specificity towards bACE2 [47]. Notably, type-1 RBDs often feature neutral LM regions, whereas type-2 RBDs commonly contain positively charged LM regions [17]. Although BANAL-20-236 possesses a type-1 RBD, it unusually contains a lysine in the LM region, which may confer an enhanced ability to bind a broader range of bACE2 molecules [56]. Monomeric BANAL-20-236-RBD was previously measured to bind bACE2$_{Ra5538}$ with a moderate affinity of 104 nM [17]; it also binds bACE2$_{R.cor}$ with a reduced affinity of 981 nM (**S11L Fig** and **S7 Table**). SARS-CoV-2-RBD is highly similar to BANAL-20-236, however, with a neutral Q in the LM region, SARS-CoV-2-RBD-Fc binding to bACE2$_{R.cor}$ was too weak to be measured by the BLI assay (**S9H Fig**). With a Q493K$_{SARS2}$ point mutation in LM, SARS-CoV-2-RBD-Fc binds bACE2$_{R.cor}$ in BLI assays (compare **S9H** and **S9L Fig**). In addition, SARS-CoV-1-RBD-Fc with an LM change (N479K$_{SARS1}$) gained weak bACE2$_{R.cor}$ binding (**S9K Fig**). These results are further consistent with the effect of the LM change (K458Q$_{Rc-o319}$) in the Rc-o319 S-protein, which considerably reduced binding to bACE2$_{R.cor}$ (**Fig 4C**). Finally, although the SARS-CoV-2, BANAL-20-236, and BANAL-20-52 S-proteins from clade1b sarbecoviruses all exhibited robust fusion activities, only BANAL-20-236 S-trimer, harbouring a lysine in the LM region, showed detectable binding to bACE2$_{R.cor}$ in BLI assays (**S11E Fig**). These observations collectively suggest that a positive-charged LM region contributes to bACE2$_{R.cor}$ receptor affinity and ultilization.

The Rc-o319-RBD:bACE2$_{R.cor}$ complex structure reveals that Rc-o319-RBD employs a featured beta-loop (BL) structure to bind to the bACE2$_{R.cor}$ molecule using a previous unobserved interaction mode that involves interactions with bACE2$_{R.cor}$ residues Lys27, Asp31, and Glu75 (**Fig 3A** and **3F**). BL change in Rc-o319-RBD-Fc is even more detrimental than the LM change, rendering bACE2$_{R.cor}$ binding completely undetectable (compare **Fig 4B** and **4C**). Notably, Lys27 is uniquely present in bACE2$_{R.cor}$ and bACE2$_{R.ferr}$ among related ACE2 molecules. However, whereas most ACE2 molecules contain a Glu75 residue, bACE2$_{R.ferr}$ instead contains a Lys75 (S17 Fig), which would abolish the K454$_{Rc-o319}$-Glu75 salt bridge observed in our Rc-o319-RBD:bACE2$_{R.cor}$ complex structure (**Fig 3A** and **3F**) and likely introduce electrostatic repulsion with the BL residue K454$_{Rc-o319}$. The specific interaction pattern between Rc-o319-RBD-BL and bACE2$_{R.cor}$ demonstrates an adaptation of the Rc-o319 RBM region to its cognate bACE2$_{R.cor}$ receptor.

While this manuscript was in preparation, Kosugi et al. reported a cryo-EM structure of the Rc-o319-RBD:bACE2$_{R.cor}$ complex [57], corroborating the interactions we observed and identifying Y463$_{Rc-o319}$ and S465$_{Rc-o319}$ in the AL region as critical for bACE2$_{R.cor}$ binding. Importantly, beyond these AL region interactions, we show that the BL and LM regions also govern bACE2$_{R.cor}$ engagement, and that replacing the Rc-o319 BL with the SARS-CoV-2 LL sequence has a substantially greater impact on bACE2$_{R.cor}$ binding than multiple LM mutations (**Fig 4B** vs 4C and see **S9D Fig** for the K458A$_{Rc-o319}$ substitution), establishing the BL as a principal determinant of high-affinity binding in Rc-o319 S-protein.

Consistently, swapping the BL with the LL of SARS-CoV-2 (the BL change) alone enabled detectable hACE2-mediated cell-cell fusion activity, even though hACE2 binding by BL variants remained undetectable using RBD-Fc (**Fig 4B**) or only very weak using S-trimer (**S10B Fig** and **S5 Table**) in BLI assays. Combined BL and RBM-loop substitutions enabled Rc-o319 RBD-Fc binding to both hACE2 and bACE2$_{R.cor}$ to be detected by BLI, albeit with very weak affinities (**Fig 4B** and **4J, and S3 Table**). These results highlight the importance of the BL/LL region together with its corresponding RBM-loop structure in ACE2 recognition, as well as the adaptation of the Rc-o319 BL/RBM-loop structures to bACE2$_{R.cor}$. Cell-cell

fusion assays showed that the BL change alone was sufficient to confer hACE2-mediated fusion, whereas the RBM-loop change alone was insufficient (**S8**B and **S8C Fig**), indicating that changes within the RBM BL region constitute the primary determinants enabling hACE2 utilization (**Figs 4** and **S8**).

Our data further show that the specialized Rc-o319 BL loop co-adapts with the Rc-o319 RBM-loop to confer high-affinity binding to its cognate $bACE2_{R.cor}$ receptor. This co-adaptation renders the RBM-loop incompatible with the SARS-CoV-2 LL region (**S7C** and **S7F Fig**), preventing an introduced LL region from mediating high-affinity hACE2 binding in BLI assays (compare **Fig 4F** and **4G**, compare **Fig 4I** and **4K**). Conversely, changing LL to the Rc-o319 BL in SARS-CoV-2-RBD completely abolishes binding to hACE2 and $bACE2_{R.cor}$ (**S12**A and **S12E Fig, and S8 Table**), likely due to incompatibility between the introduced Rc-o319 BL and the SARS-CoV-2 RBM-loop. These observations further suggest that the Rc-o319 BL and RBM-loop are coevolved structures adapted for $bACE2_{R.cor}$ binding, and their unique structural features and interactions engaged support classifying the Rc-o319-RBD as a novel type, featuring a specialized BL structure in place of the LL characteristic of previously studied type-1 RBDs. Notably, our data show that the specialized BL and RBM-loop of the Rc-o319 RBM severely limit its capacity for high-affinity hACE2 binding, with extensive mutations and insertions in these regions likely required to achieve the binding affinity necessary for cross-species transmission. Interestingly, although the structural and interaction features warrant classification of the Rc-o319 RBD as a distinct type, the BL loop generated by deletion is functionally reminiscent of the LL loop in type-1 RBDs, mediating key interactions with ACE2 despite comprising fewer residues.

Finally, our results show that, in contrast to the highly efficient utilization of $bACE2_{R.cor}$ by the Rc-o319 S-protein, $bACE2_{R.cor}$ supports only limited usage by selected clade 1 sarbecovirus S-proteins with type-1 RBDs. Notably, most of these S-proteins are from viruses that belong to the SARS-CoV-2 lineage (clade 1b), highlighting this lineage's extraordinary ability to utilize a broad range of ACE2 orthologs. Although multiple S-proteins were capable of mediating $bACE2_{R.cor}$-dependent cell-cell fusion, most exhibited little or no detectable $bACE2_{R.cor}$ binding in BLI assays, suggesting that the Rc-o319 S-protein has evolved a uniquely specialized specificity and affinity for $bACE2_{R.cor}$. Further, although structural analysis shows that the Asn38-linked glycan of $bACE2_{R.cor}$ forms specific interactions with Rc-o319 RBD residues $F423_{Rc-o319}$, $Y426_{Rc-o319}$, $H444_{Rc-o319}$, and $N459_{Rc-o319}$, our data indicate that this glycan generally dampens $bACE2_{R.cor}$ utilization by sarbecovirus S-proteins, including for Rc-o319, likely through steric effects that reduce S-protein binding. Introduction of an Asn38-linked glycan into $bACE2_{Ra9479}$ also dampened its usage by SARS-CoV-1 and BtKY72 S-proteins. We noted that, in cell-cell fusion assays, the Asn38-linked glycan had a greater impact on S-proteins mediating weak to intermediate fusion activity than on those with strong fusion activity (**Fig 5**). We speculate that this glycan may exert a more pronounced restrictive effect on viruses with limited entry capacities. Interestingly, the Asn38-linked glycan is also positioned near the binding interface of HKU5-like (**S18 Fig**), ACE2-utilizing merbecovirus S-proteins and may therefore modulate their receptor binding [58,59].

In summary, the complementary interactions between the specialized RBM, particularly the BL region, of the Rc-o319 S-protein and the corresponding interacting residues in $bACE2_{R.cor}$ appears to reflect a highly specific virus-host co-adaptation (**S17 Fig** and **S10 Table**). To enable such adaptation, the RBM of Rc-o319 has diverged substantially from that of the SARS-CoV-2 lineage, to which it is most closely related, resulting in a highly restricted ACE2 specificity. Notably, the RBM is also a major target of immune pressure, as demonstrated by the ongoing evolution of SARS-CoV-2 [60–64]. Even though certain SARS-CoV-2 lineage viruses can use $bACE2_{R.cor}$, we still do not know what evolutionary forces, perhaps immune pressure, pushed Rc-o319 S toward specialization for its host receptor.

## Limitations of the study

A limitation of this study is that the biophysical BLI assays employed relatively low-valency analytes (monomeric RBD, dimeric ACE2, or trimeric S-protein), thereby limiting avidity-mediated binding amplification and sensitivity to weak interactions. In contrast, cell-cell fusion and pseudovirus entry assays (**Figs 4**, **5, and S16**) involve highly multivalent interactions between

multiple surface-displayed S-trimers and ACE2 molecules, resulting in substantially enhanced avidity and sensitivity. Accordingly, several interactions that were undetectable by BLI nonetheless supported detectable fusion or entry in cell-based assays (**Figs 4B**, **4F**, **4G**, **4I**, **4K**, **4L, and** **5**). A further limitation is that, in the cellular context, phenotypes can be influenced by S-protein-related factors beyond receptor binding, potentially including S-protein stability, efficiency of surface expression, incorporation into pseudovirus particles, and activation by cellular proteases. This is exemplified in Fig 5, where the SARS-CoV-2 S-protein exhibited the strongest fusion activity mediated by both WT bACE2$_{R.cor}$ and the glycan mutant bACE2$_{R.cor}$-ΔGLC$_{38}$, despite no detectable binding to either receptor in BLI assays. An additional example is shown in **S16 Fig**, where a SARS-CoV-1 S-protein variant carrying the Rc-o319 RBM (SARS1$_{Rc-o319RBM}$) failed to mediate bACE2$_{R.cor}$-dependent fusion despite detectable expression, suggesting incompatibility between the swapped RBM and the S-protein backbone. While cell-based assay readouts often scale with large differences in receptor-binding affinity, they may not scale linearly when affinity differences are modest and the above-discussed S-protein-related factors are involved. Several RBM-modified S-proteins exhibited modestly increased ACE2 binding in BLI assays but reduced fusion or entry activity in cell-based assays (**Fig 4A** vs **4D** and **4G** vs **4H**). We speculate that RBM changes may affect S-protein stability, surface presentation, or interactions with host factors, leading to nonlinear relationships between receptor-binding affinity and cellular phenotype. The effects observed in cell-based systems may not fully recapitulate authentic viral infection, where additional biological constraints and host factors come into play. Addressing this will require studies with authentic viruses and in vivo models to evaluate how receptor-affinity changes influence infection in a physiological context. Despite these limitations, our findings delineate key barriers to receptor adaptation and provide a conceptual framework for assessing cross-species transmission risk.

## Materials and methods

### Protein constructs

S gene of Rc-o319 (Taxonomy ID: 694009) was codon-optimized and synthesized by Sangon Biotech (Shanghai, China). The transmembrane domain and C-terminal end of Rc-o319 (AA 1176-1235) were removed before the C-terminus of the coding sequence was fused to a T4 fibritin trimerization foldon, an HRV3C cleavage site, an Octo-His tag, and a double strep-tag to allow secretory expression and purification.

*Rhinolophus cornutus* (*R.cor*, Genebank: BCG67443.1), human (hACE2, Genebank: BAB40370.1), *Rhinolophus ferrumequinum* (*R.ferr*, Genbank: BAH02663.1) and greater mouse-eared bat *Myotis lucifugus* (*M.luci*, Genbank: XP_023609438.1) ACE genes were synthesized. Pangolin (Pangolin, Genbank: XP_017505752.1), *Rhinolophus affinis* 9479 (*Ra9479*, Genbank: QMQ39227), *Rhinolophus affinis* 787 (*Ra5538*, Genbank: QMQ39222), *Rhinolophus sinicus* WJ2 (*Rs*-WJ2, Genbank: QMQ39202.1), *Rhinolophus sinicus* 1434 (*Rs*1434, Genbank: QMQ39216.1), *Rhinolophus sinicus* 1438 (*Rs*1438, Genbank: QMQ39203.1), *Rhinolophus sinicus* 1446 (*Rs*1446 Genbank: MT394194.1), *Rhinolophus sinicus* 3357 (*Rs*3357 Genbank: AGZ48803.1), *Rhinolophus sinicus* 3359 (*Rs*3359 Genbank: QMQ39211.1), *Rhinolophus sinicus* 3366 (*Rs*3366, Genbank: QMQ39215.1) and *Rhinolophus sinicus* 5720 (*Rs*5720, Genbank: MT394182.1) ACE2 gene sequences were from a previous study [65]. The peptidase domain and the C-terminal collectrin-like domain of ACE2, bearing a carboxy-terminal Hexa-His tag, was generated for hACE2. Due to the presence of the C-terminal collectrin-like domain, these his-tagged ACE2 (AA 1-731) proteins form dimers. Additionally, the same length of ACE2s were fused to the Fc domain of human IgG with one thrombin site and a GGGG linker in between, resulting in Fc-tagged ACE2.

Rc-o319 RBD and variants (AA 293-506), SARS-CoV-2 RBD (AA 319-541), were fused with human IgG Fc domains to generate dimeric RBD-Fc proteins.

### Protein expression and purification

Proteins were expressed by transient transfection in Expi293F cells, 1 mg of DNA was transfected into 1 L of cells using Linear Polyethylenimine (PEI) transfection reagent. After transfection, cells were cultured at 33 °C for 4 days for S-proteins, 5 days for RBD and ACE2 proteins. For his-tagged S-proteins and RBD proteins, the supernatants were

harvested by centrifugation and 25 mM phosphate pH 8.0, 3 mM imidazole, 300 mM NaCl, and 0.5 mM PMSF were supplemented. Supernatant was recirculated onto a 5-ml Talon Cobalt column for about 3 times and the column was washed by 100 ml buffer A (25 mM phosphate pH 8.0, 5 mM imidazole, and 300 mM NaCl). Proteins were eluted using a linear gradient with increasing concentration of Buffer B (25 mM phosphate pH 8.0, 500 mM imidazole, and 300 mM NaCl). Eluted protein samples were quality checked by SDS-PAGE, concentrated and buffer-exchanged into PBS using a 100 kDa (for S-proteins) or 10 kDa (for RBD proteins) MWCO Amicon Ultra filtration device (Merck Millipore). The concentrated proteins were frozen in liquid nitrogen and stored at −80 °C.

For RBD-Fc and ACE2-Fc proteins, 5 days post-transfection, each culture supernatant was cleared by filtering through a 0.45 µm filter membrane before being loaded onto a 5 mL HiTrap Protein A column (Cytiva). The column was washed by PBS (10 mM $Na_2HPO_4$ and 1.8 mM $KH_2PO_4$ pH 7.4, 137 mM NaCl, and 2.7 mM KCl). Proteins were eluted using a linear gradient with increasing concentration of a citrate buffer (0.1 M citric acid, pH 3.0). Eluted protein fractions were immediately neutralized with equal volumes of 1.0 M Tris-HCl pH 8.0 buffer. RBD-Fc and ACE2-Fc proteins were concentrated, buffer-exchanged into PBS before being frozen and stored at −80 °C.

## Biolayer Interferometry (BLI) binding analysis

Binding of Rc-o319 S-proteins or RBDs to ACE2-Fc proteins was assessed using biolayer interferometry (BLI) on an Octet RED 96 instrument (FortéBio, USA). All steps were performed at 25 °C with an orbital shaking speed of 1000 rpm. All reagents were formulated in PBS-TB buffer (PBS with 0.02% v/v Tween-20 and 0.1% w/v BSA). Before the experiments, all Protein A biosensors were pre-equilibrated in the PBS-TB buffer for 10 min. ACE2-Fc proteins (11 µg/ml) were immobilized onto Protein A biosensors (FortéBio, USA) to a response value of ~1.2 nm. After a 60 s baseline step in PBS-TB, the biosensors immobilized with ACE2-Fc proteins were submerged in the analyte (S-proteins or RBDs) solutions for 300 s to measure association, before being submerged in PBS-TB for 600 s to measure dissociation. Data were reference subtracted and analyzed in the FortéBio Data Analysis Software (FortéBio) by fitting to 1:1 or 2:1 binding models to determine $k_{on}$, $k_{off}$, and $K_D$ values as previously described [66]. Raw data and fits were plotted using GraphPad Prism 8 (GraphPad Software).

Binding of dimeric his-tagged hACE2, pangolin ACE2, and bACE2s to Rc-o319 RBD-Fc protein was assessed using the BLI assay. The RBD-Fc proteins (11 µg/ml) were loaded onto Protein A biosensors (FortéBio, USA) to a level of ~1.2 nm. After a 60 s baseline step in PBS-TB buffer, the biosensors immobilized with RBD-Fc proteins were submerged in the ACE2 protein solutions for 300 s to measure association, before being submerged in PBS-TB for 600 s to measure dissociation. Data were reference subtracted and analyzed in the FortéBio Data Analysis Software.

## Cryo-EM grid preparation and data acquisition

Cryo-EM grids were prepared with Rc-o319 S-trimer samples at concentration of 1.65 mg/ml. All grids were prepared by applying 3.0 µl of S-trimer solutions directly to glow discharged (15 mA, 30 s) holey carbon grids (Quantifoil Cu mesh 300 R1.2/R1.3) before blotting. Cryo-EM grids were blotted for 2.5 s with a blot force of 4, before being plunge-frozen into liquid ethane using a Vitrobot Mark IV (ThermoFisher) at 4 °C and 100% humidity.

For Rc-o319 S-proteins, micrographs were collected using the EPU software (Thermo Fisher) on a Titan Krios G3i electron microscope (Thermo Fisher) operated at 300 keV. Micrographs were recorded at a nominal magnification of 130,000× on a Falcon 4 direct electron detector coupled to a Selectris-X energy filter set with a slit width of 10 eV. For Rc-o319 S-protein, data were collected with a raw pixel size of 0.93 Å on the image plane. Image stacks were obtained at a dose rate of ~7.1 electrons per Å$^2$ per second with a defocus ranging from -0.8 to -2.0 µm. A total of 6255 movies were collected, each with 7.05 s total exposure time, resulting in a total dose of 50 electrons per Å$^2$.

For complex formation, wild-type dimeric Rc-o319 RBD was mixed with dimeric Fc-tagged bACE2$_{R.cor}$ at a 1.2:1 molar ratio, followed by incubation at 4 °C in the presence of 0.01% glutaraldehyde for 20 min. The glutaraldehyde was

quenched by adding Tris-HCl to a final concentration of 50 mM. Gel filtration was performed to remove excess RBD on a Superose 6 10/300 GL column (Cytiva) equilibrated in PBS buffer. Complex formation was confirmed by SDS-PAGE, and the Rc-o319-RBD:bACE2$_{R.cor}$ complex peak was collected and concentrated to prepare cryo-EM grids. 3 μL of 6.8 mg/mL the RBD-ACE2 complex was mixed with 8 mM 3-[(3-Cholamidopropyl)dimethylammonio]-2-hydroxy-1-propanesulfonate (CHAPSO) before being applied onto glow discharged (15 mA, 30 s) Ni-Au-300 mesh R1.2/1.3 grids. Grids were blotted for 2.5 s with a blot force of 5 before being plunge frozen into liquid ethane using a Vitrobot Mark IV (Thermo Fisher Scientific) at 4 °C and 100% humidity. Micrographs were collected on a Titan Krios electron microscope (Thermo Fisher) operated at 300 keV using the EPU software (Thermo Fisher). Micrographs were recorded at a nominal magnification of 130,000× on a K3 direct electron detector (Gatan) coupled to a Gatan BioQuantum energy filter set with a slit width of 20 eV (Thermo Fisher). For the complex, data were collected with a raw pixel size of 0.65 Å on the image plane. Image stacks were obtained at a dose rate of ~18.7 electrons per Å$^2$ per second with a defocus ranging from -0.8 to -2.0 μm. A total of 33954 movies were collected, each with 2.67 s total exposure time, resulting in a total dose of 50 electrons per Å$^2$.

### Image processing

For dataset processed by the CryoSPARC package [67], particles were picked before 2D classification was performed. Well featured particles from the 2D classification were subjected to *ab initio* 3D classification. Homogeneous Refinement and Local Refinement were applied to the set of particles with good features identified by the *ab initio* 3D classification. For Rc-o319 S-trimer, *ab initio* 3D classification was performed with the resolution limit restricted to 10.0 Å to minimize overfitting and emphasize large-scale conformational differences. Eight initial classes were obtained from this procedure. Three well-resolved classes were subsequently subjected to independent non-uniform refinement. The resulting refined maps were then rigid-body aligned and compared using pairwise map-map correlation analysis to quantitatively assess structural similarity among the classes. Classes yielding similar structures were combined and subjected to final refinement, resulting in two Rc-o319 S-trimer structures adopting distinct conformations.

For the Rc-o319-RBD:bACE2$_{R.cor}$ complex dataset, data processing was performed mainly using cryoSPARC v4.6.2 [67]. After patch-motion correction and CTF estimation, micrographs with CTF fit resolution above 6 Å were picked by the curate exposures job. Particles were then picked using a blob picker to generate templates for the subsequent template picker. Picked particles were extracted at 3x binning and subjected to 2D classification, *ab initio* reconstruction, and heterogeneous refinement. Particles belongs to classes with high-resolution details were selected for a final round of Topaz picker by the Topaz cross validation job. Picked particles were extracted at 3x binning and subjected to 2D classification, *ab initio* reconstruction, and heterogeneous refinement again. Highly similar classes in heterogeneous refinement showing high-resolution details were combined and re-extracted at 1.5x binning and subjected to non-uniform refinement with optimization of per-group CTF parameter analysis. The CTF-refined particles and volumes were subjected to reference-based motion correction and an extra round of non-uniform refinement to get the final dimeric structure.

Due to conformational dynamics in the dimeric RBD-ACE2 complex structure, several local regions were resolved to lower resolution. Local refinement was applied to improve the local resolution at the RBD-ACE2 binding interface. Particles were symmetry expanded, the dynamic parts were masked, before the particles were subtracted through masking. The subtracted particles were imported to perform NU-Refine and local refinement jobs.

### Model building and structure refinement

A previous determined SARS-CoV-2 S-trimer structure (PDB ID: 6ZP2) [22] was used as the starting models for the Rc-o319 S-trimer structures. Model building and adjustment were performed manually in Coot 0.98 [68]. Steric clash and sidechain rotamer conformations were improved using the Namdinator web server [69]. Final structures after additional manual adjustments were refined and validated in PHENIX-1.18.261 [70]. The SARS-CoV-2-RBD:hACE2-B$^0$AT1 complex structure (PDB ID: 6M17) [27,71] was used as the starting model for the Rc-o319 RBD:bACE2$_{R.cor}$ complex structure. The

starting model was fitted to the cryo-EM density in Chimera 1.18.0 before manual adjustments were carried out in Coot. The model was refined by multiple rounds of refinement in Namdinator and PHENIX. Data collection and refinement statistics for all the reported structures are provided in S2 Table.

## GFP-split fusion assay

The split-GFP system based cell-cell fusion assay mediated by the interaction between the S-protein and the ACE2 receptor was previously described [37]. Briefly, HEK293T cells transfected with full length S-proteins and pQCXIP-GFP1–10 were prepared as effector cells, while HEK293T cells transfected with bat ACE2 (*R.cor*, *Rs*-WJ2, *Rs*1446, *Rs*1434, *Rs*1438, *Rs*3357, *Rs*3366, *Rs*3359, *Rs*5720, *Ra*9479, *Ra*5538, *R.ferr*, and *M.luci*) or pangolin ACE2 or human ACE2 (hACE2) and pQCXIP-BSR-GFP11 served as receptor cells. 24 h after transfection, effector and receptor cells were washed and resuspended in DMEM containing 10% FBS without trypsin treatment, mixed at a 1:1 ratio, and plated at ~$2 \times 10^5$ cells per well in a 96-well plate. Fluorescence images were recorded at the indicated time points using a Nikon TS2-S-SM fluorescence microscope. The GFP area was quantified with ImageJ. S-protein expression was analyzed via flow cytometry using a SARS-CoV-2 S2 antibody (1:2500 dilution, Sino Biological, 40590-T62) and western blot using an anti-FLAG antibody (1:1000 dilution, Sino Biological, 109143-MM13). The GFP-positive area was normalized to the mean fluorescence intensity of surface S-proteins, and resulting values at different time points were plotted to calculate the area under the curve (AUC) for quantitative analysis.

## VSV-Pseudovirus entry assay

VSV pseudotypes were generated according to a published protocol [72]. Briefly, HEK293T cells were transfected with 50 µg Rc-o319 WT or variant S plasmid into a 15 cm cell culture dish. After 24h, the cells were washed twice and inoculated with VSV*ΔG-Luc at a multiplicity of infection of 0.1 for 1h. After the inoculum was removed, the cells were washed 5 times using PBS with 2% FPS and further cultured with a DMEM culture medium for 24 h. The supernatant was harvested and centrifuged at 3000 rpm to remove cellular debris, filtered through a 0.45 µm syringe filter, 10-fold concentrated and stored at –80 °C in small aliquots.

To quantify entry of pseudovirons into cells, HEK293T cells stably co-expressing TMPRSS2 and ACE2 ($2 \times 10^5$ cells in 100 µL DMEM per well) were seeded in 96-well plates and inoculated with an equal volumes of pseudotype particles. 24 h post-transduction, culture supernatants were aspirated and cells lysed in 1x cell culture lysis reagent (Promega) for 20 min at room temperature. The lysates were then transferred to white 96-well plates and luciferase activity was measured using a luciferase assay system (Promega).

## Western blot

To analyze S proteins and ACE2 expression in cells, HEK293T cells were transfected with expression vectors and incubated for an additional 24h. Then, cells were washed with ice-cold PBS, and soluble proteins were extracted with cell lysis buffer (100 mM Tris-HCl pH 8.0, 150 mM NaCl, 1% NP-40) containing 1 mM PMSF (Beyotime, ST505) and 2.5 U/mL BeyoZonase Super Nuclease (Beyotime, D7121). Non-reducing SDS-PAGE loading buffer (4×) was added to each sample immediately before boiling at 98 °C for 10 min and subjected to SDS-PAGE and immunoblotting.

To analyze S-protein incorporation into pseudotyped particles, 1 mL of each VSV pseudotype preparation was subjected to ultracentrifugation (25 000 × g, 120 min, 4 °C). The supernatant (1 mL) was carefully removed. The remaining pellet was resuspended in 20 µL of 5x SDS-sample buffer, heated at 98 °C for 10 min and then analyzed by SDS-PAGE and immunoblotting.

Equal amounts of protein samples were separated by 4–10% SDS-PAGE and transferred to PVDF membranes (Millipore). After blocking with 5% (w/v) skimmed milk in TBST buffer, the membranes were incubated with primary antibodies, such as rabbit anti-SARS-CoV-2 S2 polyclonal antibody (1:2,500 dilution, Sino Biological, 40590-T62), mouse monoclonal

antibody targeting FLAG tag (1:2,500 dilution, TransGen Biotech, HT201), beta-actin (1:5,000 dilution, TransGen Biotech, HC201) and mouse anti-VSV matrix protein (Kerafast, EB0011, 1:2,500). Horseradish peroxidase-conjugated (HRP) goat anti-mouse IgG antibody (1:10,000 dilution, Beyotime, A0216) and goat anti-rabbit IgG antibody (1:10,000 dilution, Beyotime, A0208) were used as secondary antibodies. The quantitative results of VSV-M were analyzed using ImageJ software.

## Flow cytometry

To validate surface expression of S protein and its binding affinity for ACE2 following transfection, we performed flow cytometry using a BD Accuri C6 Plus flow cytometer, and the data were analyzed with FlowJo software. HEK293T cells transfected with S-protein expression plasmids were harvested 24 h post-transfection by centrifugation (3,000 rpm, 5 min, 4 °C), resuspended in ice-cold PBS containing 1% BSA, and incubated on ice for 1 h with rabbit anti-SARS-CoV-2 S2 polyclonal antibody (Sino Biological, Cat. #40590-T62, 1:500) and/or recombinant human ACE2-Fc fusion protein (10 µg/mL). Cells were then washed twice with PBS and incubated on ice for 45 min with Alexa Fluor 488-conjugated goat anti-rabbit IgG (H + L) secondary antibody (Beyotime, A0423, 1:500) and/or APC-conjugated anti-human IgG Fc antibody (BioLegend, 410711, 1:1000).

## Statistical analysis

All statistical analyses were performed using GraphPad Prism software (Graphpad Version 10.6.1). The significance of differences between the two groups was determined with an unpaired, two-tailed Student's $t$-test. For all analyses, only a probability ($P$) value of 0.05 or lower were considered statistically significant ($P > 0.05$ [n.s, not significant], $P \le 0.05$ [*], $P \le 0.01$ [**], $P \le 0.001$ [***], $P \le 0.0001$ [****]).

## Supporting information

**S1 Fig. Phylogenetic tree constructed based on 72 sarbecovirus S-protein amino acid sequences, with MERS S-protein used as the outgroup.** The tree shows the evolutionary relationships based on sequence similarities. (TIF)

**S2 Fig. Cryo-EM data processing and structure analysis for the Rc-o319 S-trimer and Rc-o319-RBD: bACE2$_{R.cor}$ complex datasets. (A-B)** Data processing pipelines are shown for the Rc-o319 S-trimer locked-1 and locked-2 structures and Rc-o319-RBD:bACE2$_{R.cor}$ structures. 3D and 2D classification steps were used to remove contaminating particles. Two conformations were identified in the 3D classification for the Rc-o319 S-trimer. **(C-D)** Local resolution maps for the Rc-o319 S-trimer and Rc-o319-RBD:bACE2$_{R.cor}$ complex structures (top panels) and global resolution assessments by Fourier shell correlation at the 0.143 criterion (bottom panels). (TIF)

**S3 Fig. Structural differences between Rc-o319 S-trimers in locked-1 and locked-2 conformations and comparison with SARS-CoV-2 S-trimer structures of different conformations. (A-B)** Top-views of the two locked Rc-o319 S-trimer structures shown in molecular surface representation. Apex residues (S465$_{Rc-o319}$) in each S-trimer are colored red. In each S-trimer structure, the distance between apices is indicated by triangles, with the distances indicated. **(C-D)** Top-views of locked-1 (PDB: 7XTZ) and locked-2 (PDB: 7XU2) SARS-CoV-2 S-trimer structures, apex positions (T500$_{SARS2}$) are colored red, with the inter-apex distances indicated. **(E-F)** Comparison of the determined locked-1 and locked-2 Rc-o319 S1 structures (extracted from the Rc-o319 S-trimer structures) with the SARS-CoV-2 S1 structure in the closed conformation (gray, PDB: 7XU3). In the locked Rc-o319 S-trimers, the receptor-binding domains (RBDs) are positioned closer to the N-terminal domains (NTDs). **(G-H)** The S1 structures within the locked Rc-o319 S-trimers closely

resemble those observed in the corresponding locked SARS-CoV-2 S-trimers (locked-1 PDB: 7XTZ, locked-2 PDB: 7XU2).
(TIF)

**S4 Fig. Structural features of the Rc-o319 S-trimers in the two locked conformations. (A)** Comparison of the locked-1 and locked-2 S1 structures of Rc-o319 S-trimer. **(B)** Rc-o319 locked-1 S-trimer structure showing the NTD and RBD. The "inserted loop" exposed to the exterior of the S-trimer is highlighted in red. **(C)** Comparison of the fatty-acid binding pockets of the fatty-acid-bound locked-1 and the fatty-acid-unbound locked-2 Rc-o319 S-trimer structures. A notable reorientation of the $Y339_{Rc-o319}$ side-chain (highlighted by red labels) is observed between the occupied and unoccupied fatty-acid binding pockets. Due to the contraction of the fatty-acid binding pocket, the side-chains of $Y339_{Rc-o319}$ and $F351_{Rc-o319}$ in the unoccupied fatty-acid pocket would clash with the modelled linoleic acid molecule, rendering the unoccupied pocket incompatible with lipid binding. **(D)** Modelled biliverdin is shown in purple stick representation with its density (blue). Residues involved in hydrophobic, hydrogen-bonding, and cation-π interactions with the bound biliverdin are shown as sticks. **(E)** An alignment of Rc-o319 and SARS-CoV-2 N-terminal domain (NTD) amino acid sequences. Modelled biliverdin interacting residues are marked by purple stars in the alignment. N1–N5 loops are framed by boxes colored in light blue, orange, green, brown and blue, respectively. Additionally, a four-amino-acid insertion, forming the "inserted loop" in the Rc-o319 NTD by comparison with SARS-CoV-2 NTD, is framed by a red box. **(F-G)** Cryo-EM densities of Domain D and the surrounding regions in locked-1 and locked-2 conformations. In the locked-1 conformation, Domain D contains a large disordered Domain D-loop; In locked-2 conformation, Domain D is fully ordered, with the disordered Domain D-loop in locked-1 refolded into two short α-helices.
(TIF)

**S5 Fig. Representative type-1, type-2, type-3, and type III sarbecovirus RBD structures with key features highlighted. (A-D)** Structures of representative type-1, type-2, type-3, and type III RBDs are shown in the same orientation. The RBM (receptor binding motif) regions are highlighted in colors, while the conserved RBD core domains are colored in gray. Black dashed circles are shown to mark the locations of the SL (Small-loop), AL (anchor-loop), and LL (large-loop) or beta-loop to highlight structural differences in the RBMs of different types of RBDs. Red dashed ovals highlight the RBM-loop structures. **(E)** Superimposition of the type III Rc-o319-RBD and the type-1 SARS-CoV-1-RBD structures. **(F)** Superimposition of the type III Rc-o319-RBD and the type-3 RmYN02-RBD structures. **(G)** An alignment of representative types 1–3 and type-III RBM amino acid sequences. Amino acid residue numbers are shown according to SARS-CoV-2 (black, top) and Rc-o319 (pink, bottom) sequences.
(TIF)

**S6 Fig. Cryo-EM density and the associated molecular model of the dimeric Rc-o319-RBD:bACE2$_{R.cor}$ complex. (A-B)** Cryo-EM density (A) and the associated molecular model (B) of the Rc-o319-RBD:bACE2$_{R.cor}$ dimer. **(C)** Detailed dimer interface interactions in the peptidase domain (PD) loop region. **(D)** Detailed dimer interface interactions in the C-terminal collectrin-like domain (CLD) neck region. Hydrogen bonds are shown as black dashed lines, salt bridges are shown as blue dashed lines, and cation-π interactions are shown as red dashed lines. Dashed boxes in (B) indicate the locations of the interfaces shown in panels C and D.
(TIF)

**S7 Fig. Interactions between the RBM-loop and the beta-loop or the large-loop of the Rc-o319 and SARS-CoV-2 RBDs. (A-B** and **D-E)** The beta-loop-RBM-loop region of Rc-o319-RBD and the large-loop-RBM-loop region of SARS-CoV-2-RBD are shown in two different viewing angles. **(A)** $Y473_{SARS2}$ of the SARS-CoV-2 large-loop and $F456_{SARS2}$ of the RBM-loop form a pi-pi interaction. **(B)** The corresponding residues $Y447_{Rc-o319}$ of the Rc-o319 beta-loop and $W430_{Rc-o319}$ of the Rc-o319 RBM-loop also form a pi-pi interaction in a substantially different conformation. **(C)** A superposition of the

Rc-o319 beta-loop-RBM-loop and the SARS-CoV-2 large-loop-RBM-loop structures. The superposition reveals a clash (highlighted by a yellow star) between the side-chains of $Y473_{SARS2}$ and $W430_{Rc-o319}$. **(D)** In the rotated view, $E471_{SARS2}$ forms a salt-bridge with the RBM-loop residue $K458_{SARS2}$, likely stabilizing the SARS-CoV-2 large-loop. **(E)** The corresponding residues in Rc-o319, $Y445_{Rc-o319}$ of beta-loop and $S432_{Rc-o319}$ of RBM-loop are not interacting. **(F)** The superposition of the Rc-o319 beta-loop-RBM-loop and the SARS-CoV-2 large-loop-RBM-loop structures in a rotated view.
(TIF)

**S8 Fig. ACE2 utilization by wild-type (WT) and variant Rc-o319 S proteins in cell-cell fusion assays. (A-C)** Representative cell-cell fusion images captured at 6 hours post-cell-mixing are shown. Effector cells expressing WT and variant Rc-o319 S-proteins were tested against receptor cells expressing either $bACE2_{R.cor}$ or hACE2. The Bottom panel: Cell-cell fusion was quantified by assessing GFP+ areas at 2, 4, and 6 hours post-cell-mixing.
(TIF)

**S9 Fig. Binding of $bACE2_{R.cor}$ by wild-type (WT) and variant Rc-o319-RBD-Fc proteins, compared with RBD-Fc proteins from SARS-CoV-1, SARS-CoV-2, BANAL-20-52 and BANAL-20-236. (A-E)** $bACE2_{R.cor}$ binding by the WT Rc-o319-RBD-Fc protein and its different variants, including $K454Y_{Rc-o319}$ in BL region (B) $K458Q/A_{Rc-o319}$ in LM (C and D), Rc-o319 RBD RBM exchanged for SARS-CoV-2 RBM ($RBM_{SARS2}$) (E) and **(F)** SARS-CoV-2-RBD-Fc with LL exchanged for BL of Rc-o319 ($BL_{Rc-o319}$) with an extra $F456L_{Rc-o319}$ mutation in the RBM-loop. **(G-J)** $bACE2_{R.cor}$ binding by wild-type RBD-Fc proteins of SARS-CoV-1, SARS-CoV-2, BANAL-20-52, and BANAL-20-236. **(K-L)** $bACE2_{R.cor}$ binding by LM variants of SARS-CoV-1-RBD-Fc ($N479K_{SARS1}$) and SARS-CoV-2-RBD-Fc ($Q493K_{SARS2}$) proteins. **(M)** An alignment of SARS-CoV-2, BANAL-20-236, BANAL-20-52, and SARS-CoV-1 RBM amino acid sequences. Compared to the SARS-CoV-2 RBM, there are two amino-acid changes, $Q493K_{SARS2}$ and $Q498H_{SARS2}$, in BANAL-20-236 and one amino-acid change, $Q498H_{SARS2}$, in BANAL-20-52. These changes likely favor $bACE2_{R.cor}$ binding by comparison with the SARS-CoV-2 $Q493_{SARS2}$ and $Q498_{SARS2}$ residues (H-J).
(PNG)

**S10 Fig. Binding of wild-type (WT) $bACE2_{R.cor}$, its Thr40Ala (removal of Asn38-glycan) variant ($bACE2_{R.cor}$-$\Delta GLC_{38}$) and hACE2 by S-trimers of different Rc-o319 variants in BLI assays. (A-E)** Binding sensorgrams were recorded by immersing biosensors immobilized with hACE2 (first panels), WT $bACE2_{R.cor}$ (second panels), or $bACE2_{R.cor}$-$\Delta GLC_{38}$ (third panels) into three-fold serial dilutions of S-trimer solutions, with concentrations ranging from 3000 to 4.09 nM. For WT (A) and the Rc-o319 AL S protein (C), concentrations ranged from 1500 to 2.04 nM.
(TIF)

**S11 Fig. Binding of wild-type (WT) $bACE2_{R.cor}$ and its Thr40Ala (removal of Asn38-glycan) variant ($bACE2_{R.cor}$-$\Delta GLC_{38}$), by S-trimers of different sarbecoviruses in BLI and FACS assays. (A-H)** Binding sensorgrams were recorded by submerging WT $bACE2_{R.cor}$ (top panels) or $bACE2_{R.cor}$-$\Delta GLC_{38}$ (removing the glycan of Asn38 by Thr40Ala mutation, bottom panels) immobilized biosensors into 3-fold serially diluted S-trimer solutions, with concentrations ranging from 800 to 1.09 nM. **(I)** Binding of $bACE2_{R.cor}$-WT or $bACE2_{R.cor}$-$\Delta GLC_{38}$ by cell-surface expressed S-proteins as assessed by flow cytometry. $bACE2_{R.cor}$-WT-Fc or $bACE2_{R.cor}$-$\Delta GLC_{38}$-Fc protein was incubated with cells expressing S-proteins before ACE2 binding was quantified using a goat anti-human IgG-APC as the probe. **(J-K)** Binding of $bACE2_{Ra9479}$-WT and $bACE2_{Ra9479}$-$GLC_{38}$ (introducing the Asn38-glycan by the Thr40Ala mutation) by S-trimers of SARS-CoV-1 and BtKY72 as assessed by BLI assays. **(L)** Binding of $bACE2_{R.cor}$-WT by monomeric BANAL-20-236 RBD in BLI assays. Monomeric BANAL-20-236 RBD was threefold serially diluted from 3000 to 4.09 nM. Estimated $K_D$ values are shown next to their corresponding binding curves.
(TIF)

**S12 Fig. Binding of $bACE2_{R.cor}$ and hACE2 by SARS-CoV-2-RBD-Fc variant proteins incorporating Rc-o319 beta-loop and RBM-loop. (A-D)** $bACE2_{R.cor}$ binding by different variants of SARS-CoV-2-RBD-Fc, including LL swapped for

the Rc-o319 BL region (SARS-CoV-2-RBD-BL$_{Rc-o319}$-Fc) (A), swapped for the Rc-o319 BL region and RBM-loop (B), LL swapped for the Rc-o319 BL region plus lamella Q493$_{SARS2}$K mutation (SARS-CoV-2-RBD-BL$_{Rc-o319}$+LM$_{Q493K}$-Fc) (C), and the whole SARS-CoV-2 RBM swapped for the Rc-o319 RBM (SARS-CoV-2-RBD-RBM$_{Rc-o319}$-Fc) (D). **(E-H)** hACE2 binding by SARS-CoV-2-RBD-Fc variants, corresponding to those in A-D. Dimeric SARS-CoV-2-RBD-Fc proteins were immobilized on the BLI sensors and tested binding against dimeric *Rhinolophus cornutus* bat ACE2 (bACE2$_{R.cor}$) or hACE2 protein as the analyte in solution. bACE2$_{R.cor}$ and hACE2 were threefold serially diluted from 3000 to 4.09 nM. Estimated $K_D$ values are shown next to their corresponding binding curves.
(TIF)

**S13 Fig. Quantification of Rc-o319 S-proteins and SARS-CoV-2 S-protein expression using western blot and flow cytometry. (A)** Total cell S-protein expression was quantified by western blot. Lysates of cells expressing different S-proteins were probed using an anti-FLAG antibody with β-actin as the control. **(B)** Quantification of SARS-CoV-2 and Rc-o319 variant S-protein expression (left panels) and their ACE2 binding (middle panels, hACE2 binding; right panels, bACE2$_{R.cor}$ binding) using flow cytometry. **(C)** Total cell hACE2, bACE2$_{R.cor}$ and bACE2$_{R.cor}$-ΔGLC$_{38}$ protein expression was quantified using an anti-flag-tag antibody as the probe by western blot. **(D)** RBD-Fc binding by ACE2 expressing cells as detected by flow cytometry. Cells surface-expressing hACE2, bACE2$_{R.cor}$-WT, or bACE2$_{R.cor}$-ΔGLC$_{38}$ were incubated with SARS-CoV-2 or Rc-o319 RBD-Fc proteins. Binding of dimeric RBD-Fc proteins was quantified using a goat anti-human IgG-APC as the probe.
(TIF)

**S14 Fig. Binding of bACE2$_{R.cor}$ to wild-type (WT) and variant Rc-o319 RBD-Fc proteins carrying individual point mutations in the combined anchor loop (AL) change (S465T$_{Rc-o319}$+A466N$_{Rc-o319}$+H470Y$_{Rc-o319}$) or a double mutation (S465T$_{Rc-o319}$+H470Y$_{Rc-o319}$), as measured by BLI assays. (A-F)** Dimeric Rc-o319 RBD-Fc proteins were immobilized on BLI sensors, and dimeric bACE2$_{R.cor}$ was used as the analyte in solution, tested using three-fold serial dilutions ranging from 3000 to 4.09 nM. $K_D$ values estimated from each binding experiment are shown alongside the corresponding binding curves.
(TIF)

**S15 Fig. SDS-PAGE of non-reducing and reducing samples of different Rc-o319 RBD-Fc proteins (10 μg) used for BLI experiments.**
(TIF)

**S16 Fig. ACE2 utilization by wild-type (WT) or Rc-o319 RBM chimeric S-proteins (SARS-CoV-1 and BtKY72) in cell-cell fusion assays. (A-H)** The RBMs of BtKY72 and SARS-CoV-1 S-protein were replaced with that of Rc-o319, generating the chimeric SARS1$_{Rc-o319RBM}$ and BtKY72$_{Rc-o319RBM}$ S-proteins, respectively. In parallel, we also introduced an Asn38-glycan into bACE2$_{Ra9479}$ by mutation Asp38 to Asn, yielding the bACE2$_{Ra9479}$-GLC$_{38}$ construct. Representative cell-cell fusion images captured at 12 hours post-cell-mixing are shown. Effector cells expressing S-proteins were tested against receptor cells expressing either wild-type or variant of bACE2$_{R.cor}$ or bACE2$_{Ra9479}$. The bottom two panels: Cell-cell fusion was quantified by assessing GFP+ areas at 2, 6, 12 and 24 hours post-cell-mixing. **(I)** Total cell bACE2$_{R.cor}$, bACE2$_{R.cor}$-ΔGLC$_{38}$, bACE2$_{Ra9479}$ and bACE2$_{Ra9479}$-GLC$_{38}$ protein expression was quantified using an anti-flag-tag antibody as the probe by western blot. **(J)** Total cell S-protein expression of SARS-CoV-1, SARS1$_{Rc-o319RBM}$, BtKY72 and BtKY72$_{Rc-o319RBM}$ was quantified using an anti-S2-tag antibody as the probe by western blot.
(TIF)

**S17 Fig. Phylogenetic tree based on analysis of 15 ACE2 sequences.** The phylogenetic tree was generated with maximum likelihood analysis. The multi-sequence alignment was analyzed using MAFFT and the key interaction residues of sarbecovirus RBD with ACE2 were identified. ACE2 interacting RBM residues of SARS-CoV-2 and Rc-o319 are

shown above and below the ACE2 sequence, respectively. Residues in the large loop (LL), lamella (LM), small loop (SL) and anchor loop (AL) regions are colored in purple, blue, orange, and green. Black lines indicate van der Waals contacts, hydrogen bonds, and salt bridges.
(TIF)

**S18 Fig. Comparison of ACE2-bound merbecovirus RBD structures with the Rc-o319 RBD-bACE2$_{R.cor}$ complex. (A)** Overview of the structural alignments between the Rc-o319 RBD-bACE2$_{R.cor}$ complex and ACE2-bound merbecovirus RBD complexes, including HKU5 RBD-*P.abramus* ACE (PDB: 9D32), HKU5-19s-*Bos taurus* ACE2 (PDB: 9E0I), MOW5-22-*P. davyi* ACE2 (PDB: 9C6O), PnNL2018B-*P.nathusii* ACE2 (PDB: 9DAK), MOW15-22-*P.nat* ACE2 (PDB: 8ZUF), NeoCoV-Bat37 ACE2 (PDB: 7WPO), and HKU5-144-2-human ACE2 (PDB: 9JJ6). **(B–D)** The Asn38-glycan of ACE2 is positioned near the interface between HKU5-like merbecovirus RBDs (HKU5 RBD, HKU5-19s, and HKU5-144-2) and ACE2.
(TIF)

**S1 Table. Kinetic parameters of different ACE2 orthologs binding to different constructs of wild-type Rc-o319 S-proteins (related to Fig 1).**
(DOCX)

**S2 Table. Cryo-EM data collection, refinement and validation statistics.**
(DOCX)

**S3 Table. Kinetic parameters of different Rc-o319 RBD-Fc variants binding to bACE2$_{R.cor}$ or hACE2 (related to Fig 4).**
(DOCX)

**S4 Table. Kinetic parameters of different sarbecovirus RBD-Fc proteins and their variants binding to bACE2$_{R.cor}$ (related to S9 Fig).**
(DOCX)

**S5 Table. Kinetic parameters of hACE2, bACE2$_{R.cor}$ or bACE2$_{R.cor}$-ΔGLC$_{38}$ binding to different Rc-o319 S-trimers (related to S10 Fig).**
(DOCX)

**S6 Table. Kinetic parameters of bACE2$_{R.cor}$ or bACE2$_{R.cor}$-ΔGLC$_{38}$ binding to different sarbecovirus S-trimers (related to S11A-H Fig).**
(DOCX)

**S7 Table. Kinetic parameters of bACE2$_{Ra9479}$-GLC$_{38}$ or bACE2$_{Ra9479}$ binding to different sarbecovirus S-trimers (related to S11J-L Fig).**
(DOCX)

**S8 Table. Kinetic parameters of different variants of SARS-CoV-2-RBD-Fc protein binding to bACE2$_{R.cor}$ or hACE2 (related to S12 Fig).**
(DOCX)

**S9 Table. Kinetic parameters of different Rc-o319 RBD-Fc variants binding to bACE2$_{R.cor}$ (related to S14 Fig).**
(DOCX)

**S10 Table. Key ACE2 residues in the ACE2-RBD interface of the orthologs tested in the cell-cell fusion assay (related to Figs 1 and S17).**
(DOCX)

## Acknowledgments

We thank the staff of the GIBH-CAS Cryo-EM Facility for their help with cryo-EM sample preparation and data collection. We thank the staff of the Guangzhou Laboratory Cryo-EM Facility for their help with cryo-EM sample data collection.

## Author contributions

**Conceptualization:** Jingjing Wang, Zexuan Li, Xinwen Chen, Xiaoli Xiong.

**Data curation:** Yong Ma, Hang Yuan, Chuanying Niu, Mei Li, Min Zhou, Wenxiu Liu, Huimin Feng, Jun He.

**Formal analysis:** Jingjing Wang, Zexuan Li, Yong Ma, Xiaoli Xiong.

**Funding acquisition:** Jingjing Wang, Yong Ma, Xinwen Chen, Xiaoli Xiong.

**Investigation:** Zexuan Li, Yifeng Teng, Xinwen Chen, Xiaoli Xiong.

**Methodology:** Jingjing Wang, Zexuan Li, Yong Ma, Zimu Li, Mei Li, Min Zhou, Wenxiu Liu, Jun He.

**Project administration:** Xinwen Chen, Xiaoli Xiong.

**Resources:** Jingjing Wang, Hang Yuan, Chuanying Niu, Banghui Liu.

**Software:** Zimu Li.

**Supervision:** Xinwen Chen, Xiaoli Xiong.

**Validation:** Jingjing Wang, Xiaoli Xiong.

**Visualization:** Jingjing Wang, Yong Ma, Xiaoli Xiong.

**Writing – original draft:** Jingjing Wang, Zexuan Li, Yong Ma, Xiaoli Xiong.

**Writing – review & editing:** Jingjing Wang, Hang Yuan, Jing Chen, Xinwen Chen, Xiaoli Xiong.

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
