## [Decision Letter · Decision Letter 0]

13 Nov 2025

PPATHOGENS-D-25-02403

Structural Basis for Sarbecovirus Rc-o319 Spike Adaptation to Rhinolophus cornutus Bat ACE2 and Constraints on Switching to Human ACE2

PLOS Pathogens

Dear Dr. Xiong,

Thank you for submitting your manuscript to PLOS Pathogens. After careful consideration, we feel that it has merit but does not fully meet PLOS Pathogens's publication criteria as it currently stands. Therefore, we invite you to submit a revised version of the manuscript that addresses the points raised during the review process.

We look forward to receiving your revised manuscript.

Kind regards,

Tyler N Starr, PhD

Guest Editor

PLOS Pathogens

Sonja Best

Section Editor

PLOS Pathogens

Sumita Bhaduri-McIntosh

Editor-in-Chief

PLOS Pathogens

orcid.org/0000-0003-2946-9497

Michael Malim

Editor-in-Chief

PLOS Pathogens

orcid.org/0000-0002-7699-2064

**Additional Editor Comments:**

Multiple reviewers noted some discrepency between cell-cell fusion and binding assays that should be clarified. The suggestion to perform pseudoviral entry assays is a good one that might help to clarify the relevant functionality of different spike:ACE2 pairs, though I would say this experiment is not entirely necessary if the authors think they can address the comments in other ways.

In the form-fields outside of the written reviews below, one reviewer noted the DURC potential of the finding of mutations that endow human ACE2 tropism to this bat sarbecovirus (though the mutations are of course found and tested in safe experimental systems). The authors could consider discussing the cost-benefit analysis that favors publication of this result in the Discussion and/or Methods.

**Journal Requirements:**

1) Please provide an Author Summary. This should appear in your manuscript between the Abstract (if applicable) and the Introduction, and should be 150-200 words long. The aim should be to make your findings accessible to a wide audience that includes both scientists and non-scientists. Sample summaries can be found on our website under Submission Guidelines:

https://journals.plos.org/plospathogens/s/submission-guidelines#loc-parts-of-a-submission

3) We notice that your supplementary Figures, and Tables are included in the manuscript file. Please remove them and upload them with the file type 'Supporting Information'. Please ensure that each Supporting Information file has a legend listed in the manuscript after the references list.

4) Please provide a detailed Financial Disclosure statement. This is published with the article. It must therefore be completed in full sentences and contain the exact wording you wish to be published.

1) Please clarify all sources of financial support for your study. List the grants, grant numbers, and organizations that funded your study, including funding received from your institution. Please note that suppliers of material support, including research materials, should be recognized in the Acknowledgements section rather than in the Financial Disclosure

2) State the initials, alongside each funding source, of each author to receive each grant. For example: "This work was supported by the National Institutes of Health (####### to AM; ###### to CJ) and the National Science Foundation (###### to AM)."

3) State what role the funders took in the study. If the funders had no role in your study, please state: "The funders had no role in study design, data collection and analysis, decision to publish, or preparation of the manuscript."

4) If any authors received a salary from any of your funders, please state which authors and which funders..

5) Your current Financial Disclosure states, "The author(s) received no specific funding for this work.".

However, your funding information on the submission form indicates recieving funds .

Please indicate by return email the full and correct funding information for your study and confirm the order in which funding contributions should appear. Please be sure to indicate whether the funders played any role in the study design, data collection and analysis, decision to publish, or preparation of the manuscript.

**Reviewers' Comments:**

Reviewer's Responses to Questions

**Part I - Summary**

Reviewer #1: In this manuscript, Wang et al. present a significant structural and functional analysis of the Rc-o319 spike protein from a Sarbecovirus identified in Japanese Rhinolophus cornutus bats. The study reveals that the Rc-o319 spike protein exhibits a highly restricted preference for its cognate bACE2R.cor due to unique features, including a novel Type-4 RBD with a distinct beta-loop (BL) structure. Through cryo-EM structures and mutagenesis studies, the authors identify molecular determinants that constrain the spike protein's ability to bind hACE2, highlighting structural limitations that restrict cross-species transmission. Additionally, comparative assays demonstrate that bACE2R.cor is compatible with only a subset of Sarbecoviruses.

Overall, this work provides critical insights into the adaptation of Rc-o319 to its bat host and evaluates its zoonotic potential, underscoring the importance of structural constraints in assessing cross-species transmission risks. Several concerns need to be addressed by the authors before the manuscript can be considered for publication.

Reviewer #2: In this study, Wang et al. determined the cryo-EM structure of Rc-o319 S-trimer and found that Rc-o319 S trimer formed two closed-conformations. The authors also determined the Rc-o319-RBD:bACE2R.cor complex structure and revealed the interaction between them. Moreover, mutagenesis experimetns elucidated that the mutation in beta loop of Rc-o319 RBM enabled to bind human ACE2. The manuscript presents an important structural analysis of the interaction between sarbecovirus spike proteins and ACE2 receptors from both human and bat. The study contributes valuable insight into the molecular determinants of cross-species receptor usage, which is highly relevant for understanding zoonotic transmission potential among coronaviruses. The paper is well structured and written, and I recommend it for publication after revision based on the comments below.

Reviewer #3: The proposed article described the structure of the Rc-o319 S-protein bound to it’s cognate ACE2 highlighting structural differences to the SARS2-human ACE2 complex. The authors have followed this up with some mutagenesis and a wider examination of bat sarbecovirus and ACE2 usage. The manuscript is nicely written with easy-to-digest figures. I have some comments and questions regarding the conclusions and novelty.

**Part II – Major Issues: Key Experiments Required for Acceptance**

Reviewer #1: 1. How were the two conformations (lock-1 and lock-2) of the Rc-o319 spike identified through 3D classification during cryo-EM data processing? Given their high similarity, is there any possibility of artificial factors influencing their identification during the classification process? Could you provide specific cryo-EM density evidence to support the distinction between these two conformations?

2. In Figures 4B, F, and G, the experimental results regarding receptor binding ability and cell fusion efficiency appear inconsistent. Since strong receptor binding is generally considered a critical step for coronavirus entry, could you clarify the reasoning behind this discrepancy? Additionally, the description of these results in the text could be further elaborated to enhance clarity and understanding.

3. In terms of binding to hACE2, is the BL loop or the RBM loop of the RBD more critical for receptor recognition? The conclusions drawn in the Results and Discussion sections regarding the relative importance of these structural elements could be more clearly articulated to better reflect the experimental findings.

4. Why does the bACE2R.cor-ΔGLC38 mutation enhance the binding affinity of Rc-o319 while simultaneously reducing its cell fusion efficiency? How do these findings shed light on the functional role of the Asn38-glycan at the interaction interface?

Reviewer #2: Major comments

1. Figure 4: The results suggest that the swapping of BL of Rc-o319 allows to use human ACE2 for cell-cell fusion, but not for ACE2 binding. On the other hand, Rc-o319 BL+SL+LM+AL+RBM-loop variant and Rc-o319-RBD-RBMSARS2 variant showed both cell fusion activity and ACE2 binding affinity. To clarify the effect of substitutions in RBM-loop and BL for ACE2 binding of Rc-o319, cell-cell fusion assay and BLI using Rc-o319 RBD BL+RBM-loop variant are recommended.

2. Figure 4: There is the explanation of the reason that Rc-o319 RBD BL variant showed cell fusion activity in human ACE2 but not binding affinity for human ACE2. However, the reason why Rc-o319 RBD AL variant showed high binding affinity for bACE2R.cor but cell fusion activity using the ACE2 was decreased is not explained in this manuscript. This explanation is recommended to be added to Discussion section.

3. In this study, not only computational modelling or cryo-EM, but also binding affinity validation using GFP-fusion assay and BLI. However, it may strengthen the manuscript to include more binding affinity validation like pseudovirus entry assay. Otherwise, the authors should discuss limitations the lack of pseudovirus entry assay.

Reviewer #3:

1. Do the types of S-proteins described in the paper group co-pylogenetically? It’s a bit confusing to read this nomenclature alongside the, also widely used, phylogenetic Clade 1-5 descriptions.

2. Is it known where in the cell the linoleic acid, oleic acid and biliverdin molecules are acquired to permit Spike locking?

3. Figure 1A: Given the results of these experiments it would be useful to understand the sequence heterogeneity in the ACE2 that is responsible for the lack of wide bat ACE2 usage and whether that is the same restriction imposed on human ACE2 binding. This could also be brought in later (see point 6 below).

4. Lines 184-186: “Because the NTD is under strong immune pressure (47, 48), these truncated and elongated loops likely endow the NTD of Rc-o319 S-protein with a distinct antigenic profile.” These references are for SARS2 where our understanding of immune selection pressure is significant. I don’t doubt there is also selection pressure on the NTD in bats but it seems a bit early to definitively state this to be the case.

5. Figure 4A-F etc. It's not clear for the described mutants if these proteins are stable. Can you include the confirmatory gels to clear this up. I can see the Western in Figure S11 but the expression seems very variable – can you just confirm equal amounts were loaded onto probes for BLI etc.

6. Figure 4: Novelty and conclusions. It’s not so surprising that a total chimera of the RBM would finally switch the functional phenotype between the cornutus and human ACE2. And the slightly contradictory results for the R.cor ACE2 glycan mutant (that its removal amplifies fusion) means it remains unclear why Rc-o319 has evolved the distinct RBM structures described. Could the authors try more mutants of other Spikes (e.g. introducing Rc-o319 changes to the SARS1 clade viruses, and seperately, comparing and mutating related Rhinophulus ACE2s (9479 and non users) to explain this in more detail). It seems to me that from a novelty perspective it would be good to ultimately understand what is it that drives the selection process.

7. Discussion lines 440: “Therefore, bACE2R.cor appears to have evolved to permit efficient binding by the S-proteins of highly specific sarbecovirus species.” I’m not sure I understand this sentence – is that positive or negative selection by the virus on its host?

**Part III – Minor Issues: Editorial and Data Presentation Modifications**

Reviewer #1: 1.It would be beneficial to provide an earlier introduction to the phylogenetic relationship between Rc-o319 and other Sarbecoviruses, as well as the diversity of RBD types, to better contextualize the significance of the findings.

Reviewer #2: Minor comments

1. Material and Method: The authors showed Western Blot data (Figure S11A and S11C). Therefore, the explanation of the method of Western Blot should be added to the section.

2. The authors should provide additional details about computational tools (e.g. versions of CryoSPARC, Chimera)

Reviewer #3: n/a see main summary for all other points.

PLOS authors have the option to publish the peer review history of their article (what does this mean?). If published, this will include your full peer review and any attached files.

Reviewer #1: No

Reviewer #2: No

Reviewer #3: No

**Figure resubmission:**
---

## [Editor Report · Decision Letter 1]

4 Mar 2026

PPATHOGENS-D-25-02403R1

Structural Basis for Sarbecovirus Rc-o319 Spike Adaptation to Rhinolophus cornutus Bat ACE2 and Constraints on Switching to Human ACE2

PLOS Pathogens

Dear Dr. Xiong,

Thank you for submitting your manuscript to PLOS Pathogens. After careful consideration, we feel that it has merit but does not fully meet PLOS Pathogens's publication criteria as it currently stands. Therefore, we invite you to submit a revised version of the manuscript that addresses the points raised during the review process.

We look forward to receiving your revised manuscript.

Kind regards,

Tyler N Starr, PhD

Guest Editor

PLOS Pathogens

Sonja Best

Section Editor

PLOS Pathogens

Sumita Bhaduri-McIntosh

Editor-in-Chief

PLOS Pathogens

orcid.org/0000-0003-2946-9497

Michael Malim

Editor-in-Chief

PLOS Pathogens

orcid.org/0000-0002-7699-2064

**Additional Editor Comments :**

Thank you for your extensive response to the initial reviewers' comments.

With respect to Reviewer 3, comment 1: please see the type I, II, III, and IV nomenclature defined by Si et al. Nat Comms 2024 (PMID 39402048). It would be better for the field as a whole to use this established nomenclature rather than introduce yet another redundant system with different relationships between “type x” and the indel structure. Si et al.'s nomenclature also has the advantage of using roman numerals which better distinguishes from the arabic numeral evolutionary clade designations.

Please address the additional editor comment from the initial decision letter:

“In the form-fields outside of the written reviews below, one reviewer noted the DURC potential of the finding of mutations that endow human ACE2 tropism to this bat sarbecovirus (though the mutations are of course found and tested in safe experimental systems). The authors could consider discussing the cost-benefit analysis that favors publication of this result in the Discussion and/or Methods.”

**Journal Requirements:**

At this stage, the following Authors/Authors require contributions: Yifeng Teng. Please ensure that the full contributions of each author are acknowledged in the "Add/Edit/Remove Authors" section of our submission form.

2) We have noticed that you have uploaded Supporting Information files, but you have not included a complete list of legends. Please add a full list of legends for your Supporting Information files (supplementary tables) after the references list.

3) Your current Financial Disclosure states, "The author(s) received no specific funding for this work.".

However, your funding information on the submission form indicates receiving funds.Please ensure that the funders and grant numbers match between the Financial Disclosure field and the Funding Information tab in your submission form. Note that the funders must be provided in the same order in both places as well.

4) Please amend your detailed Financial Disclosure statement. This is published with the article. It must therefore be completed in full sentences and contain the exact wording you wish to be published.

1) Please clarify all sources of financial support for your study. List the grants, grant numbers, and organizations that funded your study, including funding received from your institution. Please note that suppliers of material support, including research materials, should be recognized in the Acknowledgements section rather than in the Financial Disclosure

2) State the initials, alongside each funding source, of each author to receive each grant. For example: "This work was supported by the National Institutes of Health (####### to AM; ###### to CJ) and the National Science Foundation (###### to AM)."

3) State what role the funders took in the study. If the funders had no role in your study, please state: "The funders had no role in study design, data collection and analysis, decision to publish, or preparation of the manuscript."

4) If any authors received a salary from any of your funders, please state which authors and which funders..

**Figure resubmission:**
---

## [Editor Report · Decision Letter 2]

7 May 2026

Dear Professor Xiong,

We are pleased to inform you that your manuscript 'Structural Basis for Sarbecovirus Rc-o319 Spike Adaptation to Rhinolophus cornutus Bat ACE2 and Constraints on Switching to Human ACE2' has been provisionally accepted for publication in PLOS Pathogens.

Best regards,

Tyler N Starr, PhD

Guest Editor

PLOS Pathogens

Sonja Best

Section Editor

PLOS Pathogens

Sumita Bhaduri-McIntosh

Editor-in-Chief

PLOS Pathogens

orcid.org/0000-0003-2946-9497

Michael Malim

Editor-in-Chief

PLOS Pathogens

orcid.org/0000-0002-7699-2064
---

## [Editor Report · Acceptance letter]

Dear Professor Xiong,

We are delighted to inform you that your manuscript, "Structural Basis for Sarbecovirus Rc-o319 Spike Adaptation to Rhinolophus cornutus Bat ACE2 and Constraints on Switching to Human ACE2," has been formally accepted for publication in PLOS Pathogens.

Best regards,

Sumita Bhaduri-McIntosh

Editor-in-Chief

PLOS Pathogens

orcid.org/0000-0003-2946-9497

Michael Malim

Editor-in-Chief

PLOS Pathogens

orcid.org/0000-0002-7699-2064